# The microbiome of the buffalo digestive tract

Feng Tong [1,2,9], Teng Wang[3,9], Na L. Gao[3,9], Ziying Liu[2], Kuiqing Cui[2], Yiqian Duan[3], Sicheng Wu[3], Yuhong Luo[2], Zhipeng Li[2], Chengjian Yang[4], Yixue Xu[2], Bo Lin[2], Liguo Yang[5,6], Alfredo Pauciullo [7], Deshun Shi[2], Guohua Hua [5,6✉], Wei-Hua Chen [3,8✉] & Qingyou Liu [1,2✉]

Buffalo is an important livestock species. Here, we present a comprehensive metagenomic survey of the microbial communities along the buffalo digestive tract. We analysed 695 samples covering eight different sites in three compartments (four-chambered stomach, intestine, and rectum). We mapped ~85% of the raw sequence reads to 4,960 strain-level metagenome-assembled genomes (MAGs) and 3,255 species-level MAGs, 90% of which appear to correspond to new species. In addition, we annotated over 5.8 million non-redundant proteins from the MAGs. In comparison with the rumen microbiome of cattle, the buffalo microbiota seems to present greater potential for fibre degradation and less potential for methane production. Our catalogue of microbial genomes and the encoded proteins provides insights into microbial functions and interactions at distinct sites along the buffalo digestive tract.

[1] Guangdong Provincial Key Laboratory of Animal Molecular Design and Precise Breeding, School of Life Science and Engineering, Foshan University, 528225 Foshan, China. [2] State Key Laboratory for Conservation and Utilization of Subtropical Agro-Bioresources, Guangxi University, 530005 Nanning, China. [3] Key Laboratory of Molecular Biophysics of the Ministry of Education, Hubei Key Laboratory of Bioinformatics and Molecular Imaging, Center for Artificial Biology, Department of Bioinformatics and Systems Biology, College of Life Science and Technology, Huazhong University of Science and Technology, 430074 Wuhan, Hubei, China. [4] Buffalo Research Institute, Chinese Academy of Agricultural Sciences, 24-1Yongwu Road, 530001 Nanning City, P. R. China. [5] Key Lab of Agricultural Animal Genetics, Breeding and Reproduction of Ministry of Education, College of Animal Science and Technology, Huazhong Agricultural University, Wuhan, Hubei Province, China. [6] International Joint Research Centre for Animal Genetics, Breeding and Reproduction, 430070 Wuhan, China. [7] Department of Agricultural, Forest and Food Sciences (DISAFA) University of Torino Largo Paolo Braccini, 210095 Grugliasco (TO), Italy. [8] College of Life Science, Henan Normal University, 453007 Xinxiang, Henan, China. [9] These authors contributed equally: Feng Tong, Teng Wang, Na L. Gao. ✉email: huaguohua@mail.hzau.edu.cn; weihuachen@hust.edu.cn; qyliu-gene@gxu.edu.cn

Buffalo (*Bubalus bubalis*, also known as water buffalo) is a globally important domestic animal of immense value to humans, providing economic value from milk, meat, and leather production and draft power[1]. The worldwide estimated population of more than 200 million buffaloes is relied upon by more than two billion people—a greater number than for any other domesticated animals[2]. Buffaloes are even-toed, hoofed mammals of the Bovidae family, genus *Bubalus*, and tribe Bovini. There are two subspecies of the domesticated water buffalo: swamp buffalo (*Bubalus bubalis* carabanesis, $2N = 48$) and river buffalo (*Bubalus bubalis* bubalis, $2N = 50$)[3,4].

The distribution of swamp buffaloes overlaps closely with that of rice agriculture in East and Southeast Asian countries (e.g. China, Vietnam, and Thailand), where they have served as the primary draft animals for rice cultivation for thousands of years[5]. Buffaloes are herbivores, and most of their diet is composed of plant fibre such as forage grasses. They possess a characteristic four-chambered stomach (FC stomach, including the rumen, reticulum, omasum, and abomasum for short) that is specialized for transforming the forage with low nutritional value into high-quality animal protein and adapting to the life cycle of forage grass[6]. The digestibility of organic matter was greatest by buffaloes is greatest when the animals are fed hay[7].

The importance of the microbiota of different sites in the digestive tract is now well recognized and has been linked to the functions and physiology of the corresponding DT sites[8]. For example, the rumen, the first and most important stomach site in ruminants, contains abundant bacteria capable of digesting cellulose (the main component of plant cell walls), including *Fibrobacter*[9–12], *Ruminococcus*[13,14], and *Butyrivibrio*[15,16] bacteria; similarly, *Prevotella*, a group of bacteria capable of degrading noncellulose plant fibres, is abundant in the rumen[17–20]. Archaea are also abundant in the rumen and other sites of the FC stomach; however, they are the main producers of methane, a major greenhouse gas[21–23] and are thus targets for elimination[24]. In addition to their roles in food digestion and nutrient absorption, the rumen and gut microbiota have been linked to more pronounced phenotypes, such as the milk production and quality of cattle[25,26]. To date, research on the microbial ecology along the DT of ruminants has mostly focused on the rumen[27–36], along with a few studies on the gut/faeces[25,37], while other sites have been largely overlooked, despite accumulating evidence supporting their important roles. For example, in humans, the gut microbiota has been linked to many aspects of human life, including health[38,39] and diseases[40–44], development[45,46], responses to drugs and treatments[47–49]. In addition, recent studies on the buffalo DT microbiota have mostly used 16 S rRNA gene sequencing[35], while whole-genome metagenomics has been applied to only a few samples[25].

The application of metagenomic next-generation sequencing (mNGS) has greatly facilitated the reconstruction of large numbers of metagenome-assembled genomes (MAGs) in model organisms, including cows[30,36,50–52], goats[37], pigs[53,54], mice[55,56], and chickens[57–60], and revealed their associations with the health and diseases of their hosts[61,62]. In this study, we set out to fill the gaps in ruminant research by performing a comprehensive survey of the microbial ecology of the buffalo's DT. We collected in total 695 samples from eight DT sites in three sections: the FC stomach, intestine, and rectum. We performed metagenomic sequencing and obtained 4960 MAGs, which covered in total 85% of the raw sequencing reads. We identified known interactions between microbes and the rumen, where particular enrichment of fibre-digesting and methane-producing microbes was observed, in addition to novel and important interactions in other FC stomach sites and DT sections. We annotated 5,862,748 non-redundant proteins from the MAGs, many of which showed differences in abundances and were related to site-specific functions. Thus, our taxonomic and functional characterization of the microbial of the buffalo's DT provides insight into microbiota functions and interactions with distinct DT sites and will contribute to overcoming the barriers to manipulating the gastrointestinal microbiota to improve animal productivity in buffalo as well as the related ruminant animals.

## Results

**Generation and quality assessment of MAGs along the digestive tract of buffalo.** To provide a comprehensive collection of microbes associated with the digestive systems of buffalo, we collected in total 695 samples from different sites along with the DTs of buffaloes from six provinces of China (Supplementary Data 1), including 211 samples from four distinct sites of the FC stomach (rumen, reticulum, omasum, and abomasum), 85 samples from three sites in the intestine (caecum, colon, and jejunum) and 399 samples from the rectum (faeces) (Fig. 1a). The buffaloes varied in terms of their breeds (three main breeds: river buffalo, swamp buffalo, and hybrid buffalo), developmental stages (calf, breeding, and adult), and sexes (oxen and cows) (Supplementary Data 1).

We subjected the 695 samples to mNGS using the Illumina NovaSeq 6000 platform for paired-ended sequencing with a read-length of 150 bp. After removing vector and low-quality sequences and contamination from the host and food genomic sequences, we obtained a total of 11 Tb of clean data for further analyses (see Methods for details). On average, we obtained 41,842,231 pairs of clean reads and 6,244,074,222 bases from each sample.

To obtain metagenome-assembled genomes (MAGs), we adopted a customized bioinformatics analysis workflow (see Supplementary Fig. 1 for a graphical representation; Methods). Briefly, the clean reads were assembled by using metaSPAdes[63] and MEGAHIT;[64] the resulting 109,471,448 contigs were grouped into 58,094 bins using metaBAT2[65] with the default parameters. All bins were aggregated and dereplicated using dRep[66] (v.2.3.2) with the default overlap threshold (-nc 0.1) and were clustered into strain-level and species-level genome bins at 99% and 95% of the average nucleotide identity (ANI), respectively, followed by the application of CheckM[67] (v.1.0.18) for quality assessment. Finally, we obtained a non-redundant set of 4960 bins (MAGs) at the strain level with completeness ≥80% and contamination ≤10% and 3255 MAGs at the species level. Rarefaction analysis indicated that the curves could reach a plateau when using samples from the FC stomach or rectum or from sites all combined, although not when using intestinal samples, as fewer of these samples were available (Fig. 1b).

Among the resulting MAGs at the strain level, 1575 (31.75%) were high-quality draft genomes as defined by Bowers et al.[68] with ≥90% completeness, ≤5% contamination, and the inclusion of the 23 S, 16 S, and 5 S rRNA genes and at least 18 tRNAs, while 3228 (65.08%) met the score criterion defined by Parks et al.[69] (completeness − (5 × contamination) ≥50) (Fig. 1c, d). The sizes of the MAGs ranged from 402 kilobases (Kb) to 6.1 megabases (Mb), with a median of 2.1 Mb (Fig. 1f). Moreover, a similar proportion of the quality assessment results (e.g. 1214 (37.3%) and 1951 (59.9%)) corresponded to high- and medium-quality MAGs (Supplementary Fig. 2a, b), and the approximate distributions of N50 values, genome sizes, and the numbers of contigs per genome were obtained among the species-level MAGs (Supplementary Fig. 2c–e). Compared with the complete genomes in the NCBI RefSeq prokaryotic genome database (457 archaeal and 28,011 bacterial genomes at the complete and chromosomal assembly levels; downloaded as of June 28, 2021),

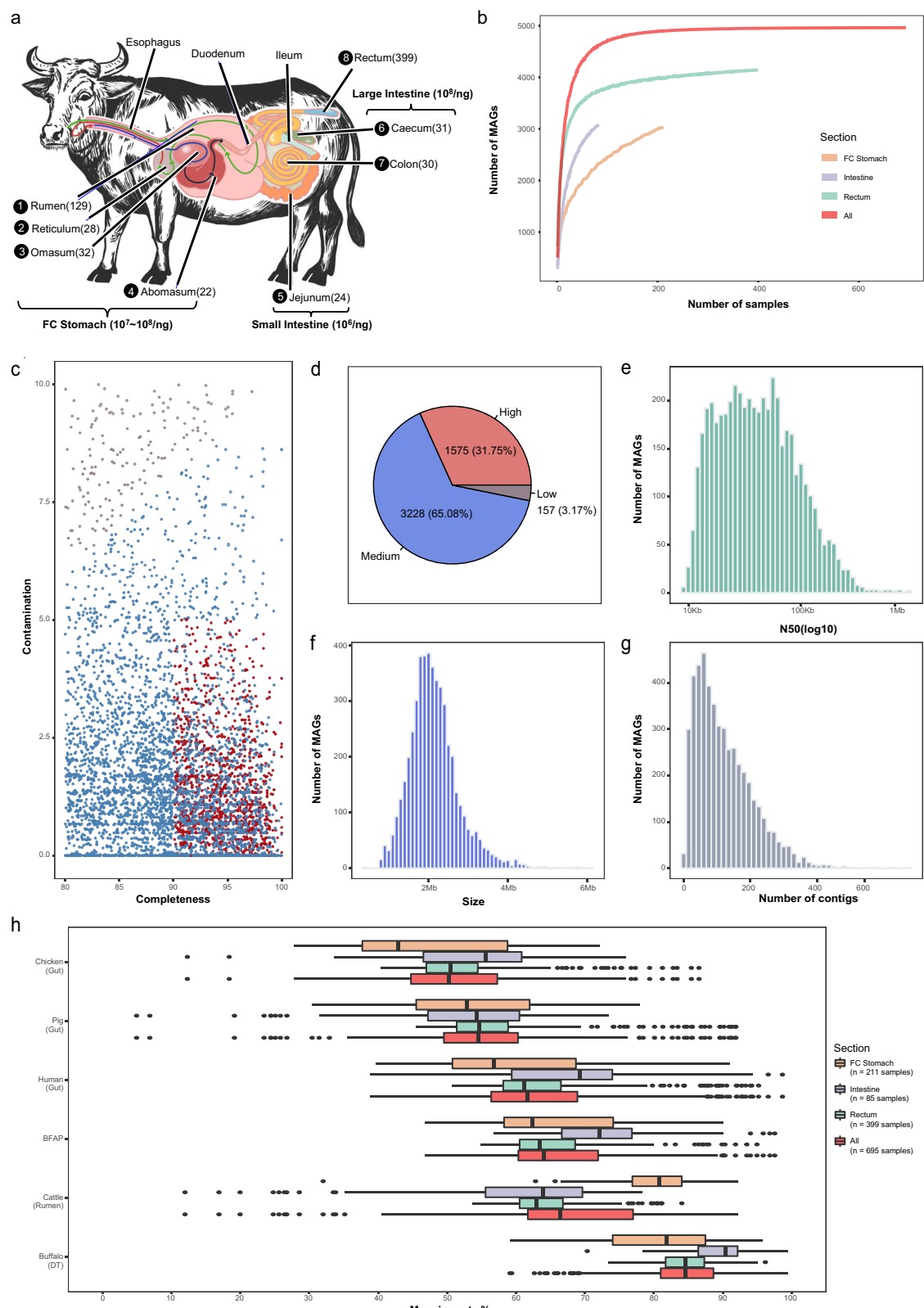

our strain-level MAGs were indeed shorter (Supplementary Fig. 3a). However, due to the lack of representativeness of buffalo microbial genomes in public databases, the differences could in part be due to the distinct characteristics of buffalo MAGs. In addition, we believe that the completeness as measured by the presence of key genes is a better index than genome sizes since the latter parameter varies significantly even within species. Hence,

we also compared the completeness and genome sizes of the buffalo MAGs with those of 4941 cattle rumen MAGs from Stewart et al.[30], and we found that although the buffalo MAGs were also shorter than the cattle rumen MAGs (Supplementary Fig. 3b), the former presented significantly higher completeness values than the latter (Supplementary Fig. 3c). These results may suggest that buffalo-associated microbes tend to have compact

**Fig. 1 Generation and quality assessment of 4960 metagenome-assembled genomes (MAGs) along the digestive tract of buffalo. a** Graphical representation of samples collected from distinct sites along the digestive tract of buffalo. Specific sites are labelled; the arrows inside the digestive tract indicate the flow of the food. The numbers in parentheses after the labels indicate the number of samples obtained for this study. The numbers before the labels indicate the order of food flow; please note that only the sites from which samples were collected are ordered. The numbers above and below the curved brackets indicate the estimated numbers of microbes per nanogram DNA in the FC stomach, small intestine, and large intestine of the DT. **b** Rarefaction curves of assembled genomes (MAGs) as a function of the input samples. Here the Y-axis represents the median number of non-redundant MAGs at the strain level (those shown in panel **c**) obtained from a specific number of samples (X-axis) via random 100 times with replacement. **c** Quality assessment (i.e. completeness and contamination statistics) of the 4960 non-redundant MAGs. Each point represents one MAG. Red points indicate the highest-quality genomes, with ≥90% completeness, ≤5% contamination, and the inclusion of the 23 S, 16 S, and 5 S rRNA genes and at least 18 tRNAs. All other MAGs showed >80% completeness and ≤10% contamination. The MAGs in blue show a quality score ≥50 as defined by Parks et al.[69], whereas those in grey show a quality score <50. **d** The pie chart shows the numbers and relative proportions of the red, blue, and grey MAGs in **c**. Histograms in **e**, **f**, and **g** show the distributions of N50 values, genome sizes, and the numbers of contigs per genome, respectively, for the 4960 MAGs. **h** Increased coverage of metagenomics reads is achieved by our MAGs relative to reference microbial genomes and MAGs of other model organisms. The percentages of reads from the FC stomach (orange, $n = 211$ samples), intestine (purple, $n = 85$ samples), rectum (green, $n = 399$ samples), and all samples (red, $n = 695$ samples) that could be mapped to the collected datasets are shown. Boxplots show the median and the 25th and 75th percentiles, the solid lines indicate the minima and maxima, and the points falling outside of the whiskers of the boxplots represent the outliers. In addition to our MAGs (buffalo), chicken, pig, and human MAGs were obtained from gut metagenomes;[53, 57, 70, 107] BFAP represents the combination of reference genome datasets including bacterial, fungal, archaeal, and genomes from the Hungate collection;[71] cattle MAGs were obtained from rumen metagenomes[30].

genomes; however, further validation is needed to verify this conclusion.

The MAGs contained 1 to 679 contigs, with a median of 102. The contig sizes of the MAGs ranged from 2.5 Kb to 1.5 Mb, with N50 values (50% of assembled bases in contigs larger than the N50 value) ranging from 5.4 kb to 1.4 Mb (Fig. 1e, g). To further access the quality of the resulting MAGs, we annotated tRNA and 16 S rRNA genes encoded by these MAGs. All MAGs encoded tRNA genes with an average of 18.6 tRNA genes per MAG; in total, 3899 of the strain-level MAGs and 3233 of the species-level MAGs encoded at least 18 tRNAs, indicating the high quality of the MAGs. However, according to the strain- and species-level MAGs, respectively, only 285 and 210 MAGs encoded at least one complete 16 S rRNA gene, while an additional 394 and 314 MAGs contained a partial 16 S rRNA gene (Supplementary Data 2); these results are consistent those of Stewart et al.[30].

To determine whether our MAGs could improve the coverage of microbial genomes associated with the buffalo DT relative to NCBI RefSeq genomes and other public datasets, we mapped all the metagenomic clean reads to the resulting strain-level MAGs and compared the overall mapping rates against the MAGs of selected organisms (human[70], chicken[57], pig[53], and cattle[30]) and reference microbial genomes from public databases (NCBI RefSeq genomes plus Hungate collection genomes[71], referred to as the BFAP dataset). A high mapping rate indicates that the majority of the microbial genomes in a sample can be represented by the target dataset, while a lower mapping rate indicates the existence of novel genomes in the sample that is not covered by the target dataset. This strategy has been extensively used in metagenomic analyses[30,72]. As shown in Fig. 1h, the buffalo raw reads showed the highest mapping rate to our MAGs with an average of 85%, followed by the cattle Rumen Uncultured Genomes (RUGs) of Stewart et al.[30], the BFAP dataset, and the MAGs of humans, pigs, and chickens.

Our FC stomach samples showed lower mapping rates to reference genomes and MAGs than the samples from the intestines and rectum (Fig. 1h), Similar results could be observed in gut microbial datasets from humans, pigs, and chickens. However, the mapping rates against the cattle dataset were an exception because this dataset consisted of MAGs from rumen metagenomes[30]. These results were consistent with the fact that FC stomach metagenomes are less well in public databases, and most metagenomic studies have focused on the gut. The overall mapping rates of the FC stomach samples against cattle and buffalo MAGs were close (Fig. 1h, orange boxes), suggesting that

their FC stomach metagenomes were similar; however, the mapping rates of the buffalo intestine and rectum samples against cattle RUGs were significantly lower (less than 65%), suggesting that gut metagenomes were underrepresented in Stewart et al. 's data[30]. Since the cattle and buffalo are closely related, we also mapped raw sequencing reads of cattle metagenomes obtained from[30] from our MAGs. On average, 71% of the cattle reads could be mapped to our MAGs, compared to 82.5% mapped to cattle RUGs (Supplementary Fig. 4).

Together, we obtained a total of 4960 strain-level MAGs and 3255 species-level MAGs indicating potential new strains and species, respectively, which significantly improved the coverage of the microbes in the buffalo digestive tract, especially those in sections of the intestine and rectum.

**Taxonomic annotation of buffalo MAGs.** We next assigned taxonomic classifications to the MAGs using the Genome Taxonomy Databasse Toolkit (GTDB-TK)[73]. Almost of the MAGs could be classified into known taxonomic categories at the higher levels, such as the kingdom, phylum, class, and order levels; however, at more refined levels especially the species level, only 460 (9.3%) of the strain-level MAGs and 415 (12.7%) of the species-level MAGs could be classified as known species (Fig. 2a and Supplementary Fig. 2f), indicating that most of the MAGs were novel (i.e. not present in GTDB-TK). Bacteria and archaea showed similar classification results among the strain- and species-level MAGs (Fig. 2b, c and Supplementary Fig. 2g). For the following analysis, we used the strain-level MAGs unless stated otherwise.

As shown in Fig. 2, the dominant phyla according to the numbers of MAGs that could be assigned were Bacteroidota ($n = 2200$) and Firmicutes_A ($n = 1736$) (dominated by classes Bacteroidia and Clostridia), followed by Verrucomicobiota ($n = 191$), Firmicutes ($n = 172$), Spirochaetota ($n = 164$), and Proteobacteria ($n = 147$). The dominant orders included Bacteroidales ($n = 2194$) and Oscillospirales ($n = 1101$), while the dominant families included Bacteroidaceae ($n = 676$), CAG-272 ($n = 588$), Rikenellaceae ($n = 483$), and UBA932 ($n = 368$). At the genus level, the dominant groups included *Alistipes*, *RC9*, and *Prevotella* (Supplementary Fig. 5). Seventy-four archaeal MAGs belonging to three phyla, Halobacterota ($n = 42$), Thermoplasmatota ($n = 20$), and Euryarchaeota ($n = 12$), were identified. The top three genera included *Methanocorpusculum* ($n = 27$), *Methanomicrobium* ($n = 13$), and *Methanobrevibacter_A* ($n = 11$), all of which are methanogens.

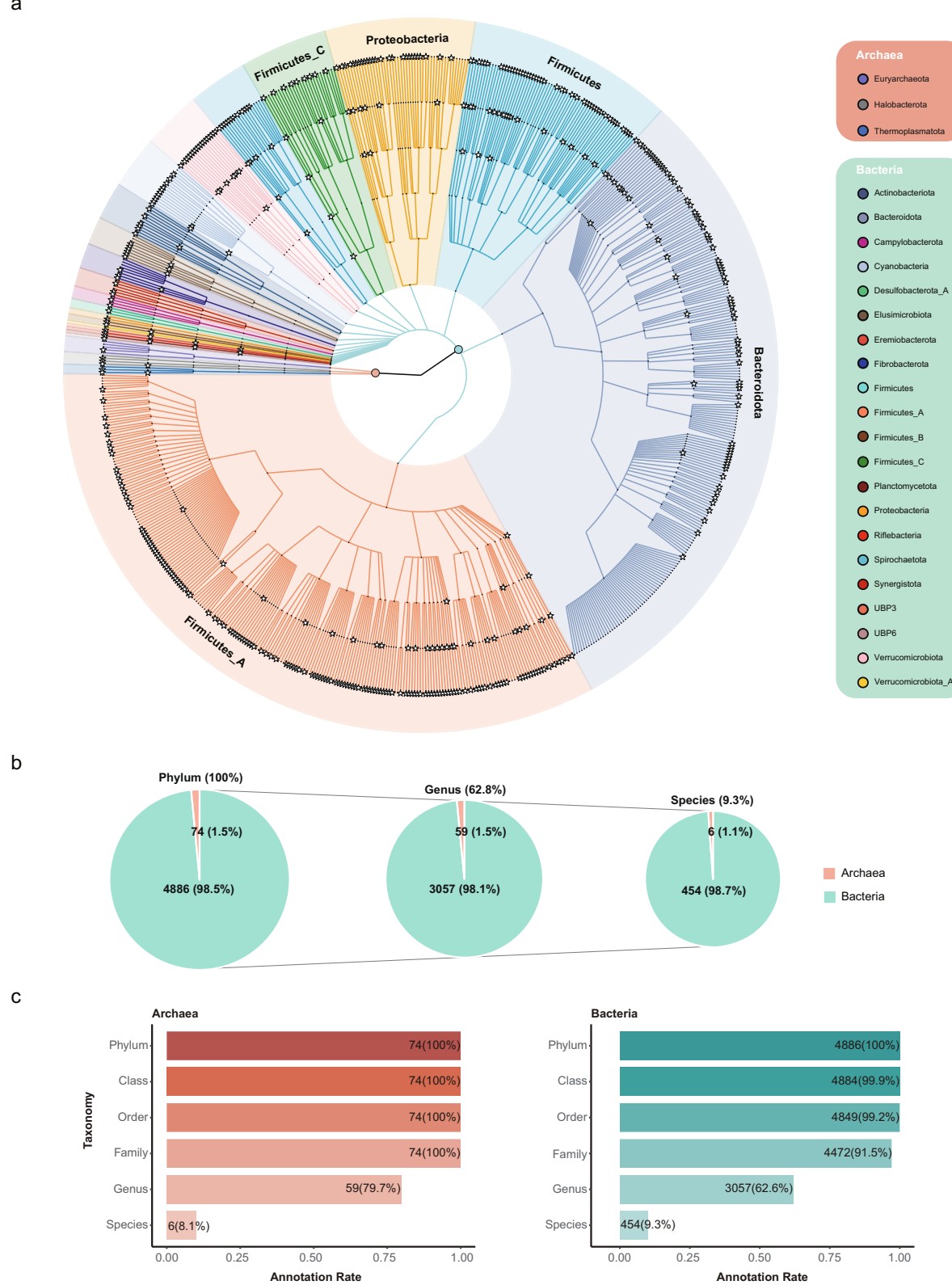

**Taxonomic characteristics of MAGs in different sections along the digestive tract**. To investigate the distributions of the 4960 MAGs in different sections along with the DT and their putative interactions with the host, we first determined the coverage and relative abundances of all the MAGs in each sample. We mapped the clean reads of a sample to all MAGs and calculated the coverage of a MAG as the total aligned bases divided by the total

bases of the MAG[74], and the relative abundance of a MAG as the percentage of reads mapped to the MAG among the total reads mapped to all MAGs (see Methods for details).

Using a coverage 1X coverage as the cut-off of presence/absence, we found that 3032, 3081, and 4141 MAGs were present in at least one sample from the three sections (Fig. 3a). According to this criterion, we found that 1692 (34% out 4960) MAGs were

**Fig. 2 Classification of the MAGs into known groups at different taxonomic levels. a** Phylogenetic relationships and taxonomic classifications of the 4960 strain-level MAGs from the buffalo digestive tract. A circular cladogram representation of the phylogenetic relationships of the 4960 MAGs is shown here. The stars at the internal and leaf branches indicate novel branches that were not present in GTDB-TK[73]. Phyla are highlighted with different background colours, with the outermost labels indicating selected phyla with the greatest numbers of MAGs. The right panel lists all of the phyla of archaea and bacteria; the fill colours of the dots before the phylum labels correspond to the branch colours in the phylogenetic tree. **b** Classification rates of 4960 MAGs at different taxonomic levels. The numbers above the pie charts indicate the percentages of the MAGs (out of 4960) that could be annotated at the respective levels; the numbers inside the pie charts indicate the percentages of archaea (orange) and bacteria (green) in each pie. **c** Classification rates of archaea (left) and bacteria (right) at different taxonomic levels. The numbers indicate the numbers of MAGs classified.

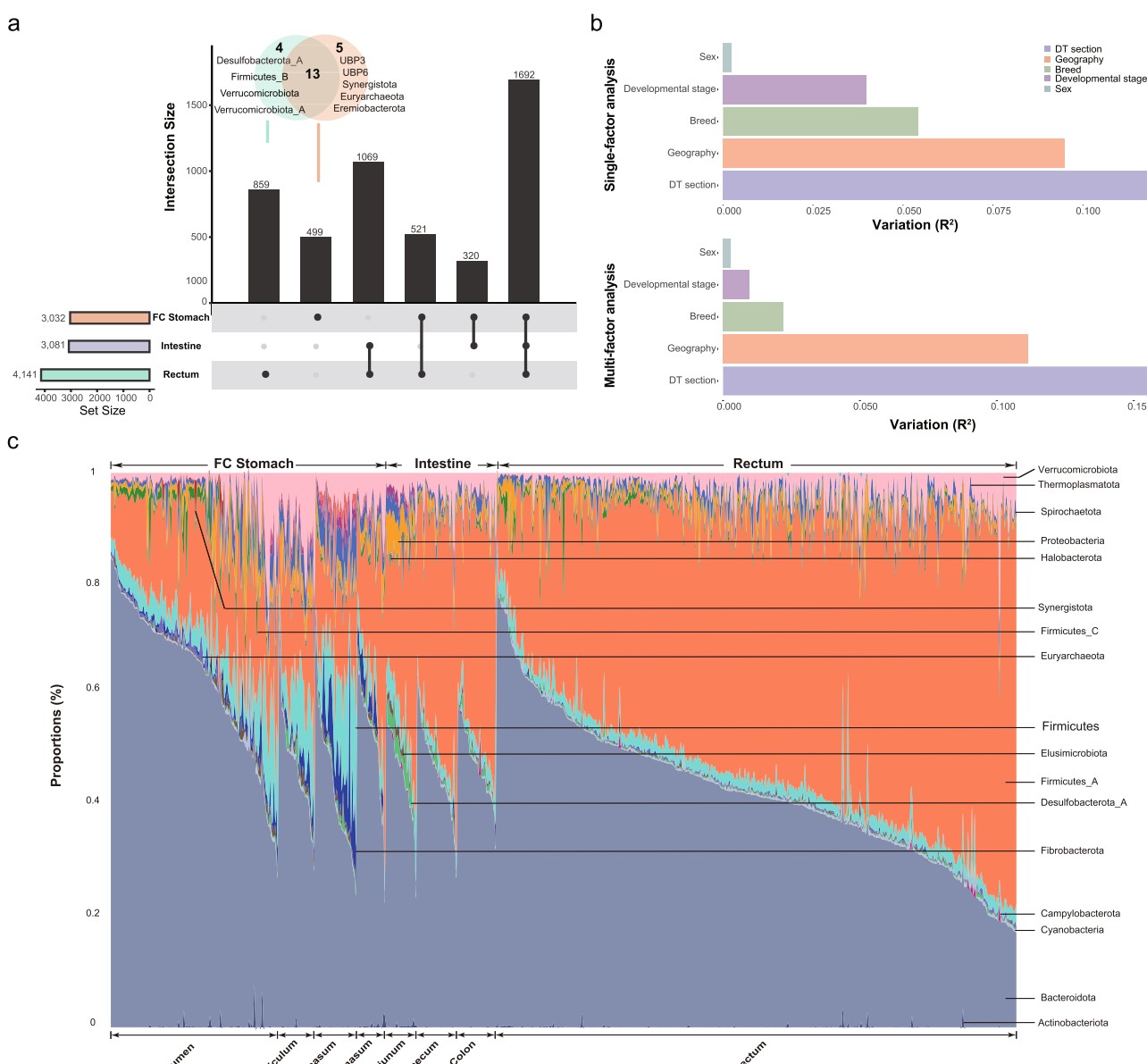

**Fig. 3 Taxonomic characterization of buffalo MAGs along the digestive tract (DT). a** Distributions of MAGs in different sections along the DT. Here, a MAG is considered to be present in a section if its coverage is greater than 1 × in more than one sample from section. The coverage is defined as the total bases mapped to a MAG in a sample divided by its length. **b** Bar plot summarizing permutational multivariate analysis of variance (PERMANOVA) analyses results using single and multiple factors. All factors were found to be significantly associated with gut microbial variations ($P = 0.001$). The variations were derived from between-sample weighted UniFrac distances. The pairwise Wilcoxon rank-sum test was used to compare groups. The bars are coloured according to metadata categories. **c** Stream graph displaying of phyla along the buffalo DT. The *X*-axis indicates the samples clustered by the sampling sites along the DT. The *Y*-axis indicates the relative abundances of the phyla in each sample.

present in all three sections, 1910 MAGs (38.5%) were present in samples from two sections, and only a small proportion of the MAGs was section specific (499 in FC stomach and 859 in rectum; Fig. 3a). We did not find intestine-specific MAGs according to this criterion, likely because the intestine is positioned between the FC stomach and rectum (Figs. 1a and 3a). At the phylum level, five phyla were FC stomach specific, among which three (UBP3, UBP6, and Eremiobacterota) were designated candidatus, indicating that thet belonged to unknown phyla. Regarding the other two phyla, Euryarchaeota consists of species related to methane metabolism, while Synergistota consists of oral bacteria in humans, was first identified in the goat rumen, and is usually associated with toxic compound degradation[75]. The main rectum-specific was Verrucomicrobiota, which is found in soil, water, and faeces[76].

We further analyzed the relative abundances of the MAGs along with the DT and found significantly different alpha diversities among all sections, measured according to both the Shannon and Simpson indexes (Supplementary Fig. 6). The rectum presented the highest diversity, followed by the FC stomach and intestine (Supplementary Fig. 6), which was partially consistent with the numbers of MAGs identified in the three sections (Fig. 3a). We then applied principal coordinates analysis (PCoA) to reveal the Bray-Curtis distances among the samples and found that the overall microbial profiles of the intestine and rectum were similar ($R^2 = 0.048$; $P = 0.001$, pairwise nonparametric multiple analysis of variance (MANOVA)), while both were significantly different from that of the FC stomach ($R^2 = 0.192$, $0.270$, $P = 0.001$, $0.001$, pairwise nonparametric MANOVA; Supplementary Fig. 7a). Interestingly, samples from the jejunum of the intestine formed their own cluster and were distant from the samples from other DT sites (Supplementary Fig. 7b), indicating a distinctive microbial structure at the site; this was also consistent with the alpha diversity of the jejunum, which was different from those of other parts of the DT (Supplementary Fig. 6).

To maximize the discovery of DT-associated microbial genomes, we collected samples from animals that differed in their geographical locations, breeds, sexes, and developmental stages, which are factors known to affect the microbial compositions. We thus also investigated the effects of these host and environmental factors on the microbiota using permutational multivariate analysis of variance (PERMANOVA) implemented in the R package 'vegan'. As expected, the DT sections exerted the strongest effects in both the single- and multifactor analyses, followed by geography, breed, developmental stage, and sex (Fig. 3b). Since the sampling strategy applied in this study was optimized to compare microbial compositions across DT sections, we focused on the comparative analysis between DT sections and left the comparisons of other factors for future studies.

**Distinctive patterns of MAGs along the digestive tract coincide with their functions**. As shown in Fig. 3e, the overall microbial structures of the FC stomach, intestine, and rectum were different from each other (Fig. 3e), with Firmicutes and Bacteroidota being the two most abundant phyla. Firmicutes_all, including Firmicutes, Firmicutes_A, Firmicutes_B, and Firmicutes_C according to GTDB, accounted for 79%, 85%, and 90% of the total microbial abundances in three DT sections on average. Interestingly, Bacteroidota showed a decrease in abundances along the digestive tract (Fig. 4a), while Firmicutes_all showed the opposite pattern (Fig. 4b). Consequently, the Firmicutes to Bacteroidota ratio (F/B ratio, Fig. 4c) was lowest in the FC stomach and highest in the rectum. Previous results have linked an increased F/B ratio with an increased capacity for harvesting energy from the diet[77,78],

consistent with the physiological roles of the three sections. In addition, the F/B ratio in the rumen has been linked to the milk fat yield in cows[79–81].

The decreasing abundances of Bacteroidota along the DT were partly due to lower counts of *Prevotella*, the main genus of Bacteroidota that was highly abundant in the FC stomach, especially in the rumen, whereas its counts were significantly lower in other sections and DT sites (Fig. 4h); this genus accounted for 33.1% of the total microbial abundances in the rumen. *Prevotella* species are associated with noncellulose plant fibre degradation and are the largest single bacterial group reported in the rumen of cattle and sheep under most dietary regimes[82]. Most of the species in this genus are unclassified, including the most abundant species (Supplementary Fig. 8). In addition to the role of *Prevotella* species in plant degradation, they also played an important role in preventing ruminal acidosis[82–84].

Cellulose is the main component of the cell wall of plants, and its degradation by DT-associated microbes is crucial in buffalo. *Fibrobacter*, *Ruminococcus*, and *Butyrivibrio*[85] have generally been considered as the main microbes responsible for cellulose digestion (the substrates of Buyrivibrio are diverse, including cellulose, hemicellulose, and proteins). We thus explored their distributions along with the DT. As expected, *Fibrobacter_all* (including *Fibrobacter* and *Fibrobacter_A* according to GTDB; Fig. 4e), *Ruminococcus_all* (including *Ruminococcus_E*, *Ruminococcus_A*, and *Ruminococcus* according to GTDB; Fig. 4f), and *Butyrivibrio_all* (including *Butyrivibrio_A* and *Butyrivibrio* according to GTDB; Fig. 4g) were all significantly abundant in the FC stomach[86,87] and were less abundant in other DT sites (Fig. 4e–g). Among these taxa, *Fibrobacter_all* was the most abundant. Our results showed that the total abundances of *Fibrobacter_all* were significantly higher in all four FC stomach sites (i.e. rumen, reticulum, omasum, and abomasum) than in the other two sections; however, we found the highest abundance of *Fibrobacter_all* in the omasum rather than in the rumen (widely believed to the site with the highest abundance of this group), indicating that the omasum may play important roles in cellulolytic digestion.

All of the archaeal species that we identified were methanogens, and they were highly abundant in the FC stomach and intestine (Fig. 4d). These results contradicted the current understanding that only the FC stomach alone, especially the rumen, is the main organ of methane metabolism[88] and highlight the importance of the intestine in methane metabolism. Strikingly, close examination revealed that the total abundance of these methanogens peaked in the omasum (Fig. 4d), coinciding with *Fibrobacter_all* (Fig. 4e). Interestingly, archaea and *Fibrobacter_all* showed significant correlations in their overall abundances in both the FC stomach and intestine (Fig. 4i). In addition, we performed correlation analysis between all bacterial genera and archaea (Supplementary Data 3), and found that *Fibrobacter_all* showed the highest correlation ecoefficiency with the methanogens (Supplementary Fig. 9). These results strongly suggested that *Fibrobacter_all* may play important roles in methane production. Previous studies showed that methanogens in the rumen potentially affect the metabolism of microbes that degrade cellulose by utilizing hydrogen, and our results suggested that they play similar roles in the omasum. Together, these results highlighted the important roles of the omasum in both methane metabolism and cellulose degradation and the possible functional link between these two processes[89].

In total, we identified 359 differentially abundant taxa at the genus level using the linear discriminant analysis effect size (LEfSe)[90] with linear discriminant analysis (LDA) scores >2. Among these taxa, 172, 42, and 145 showed significantly higher

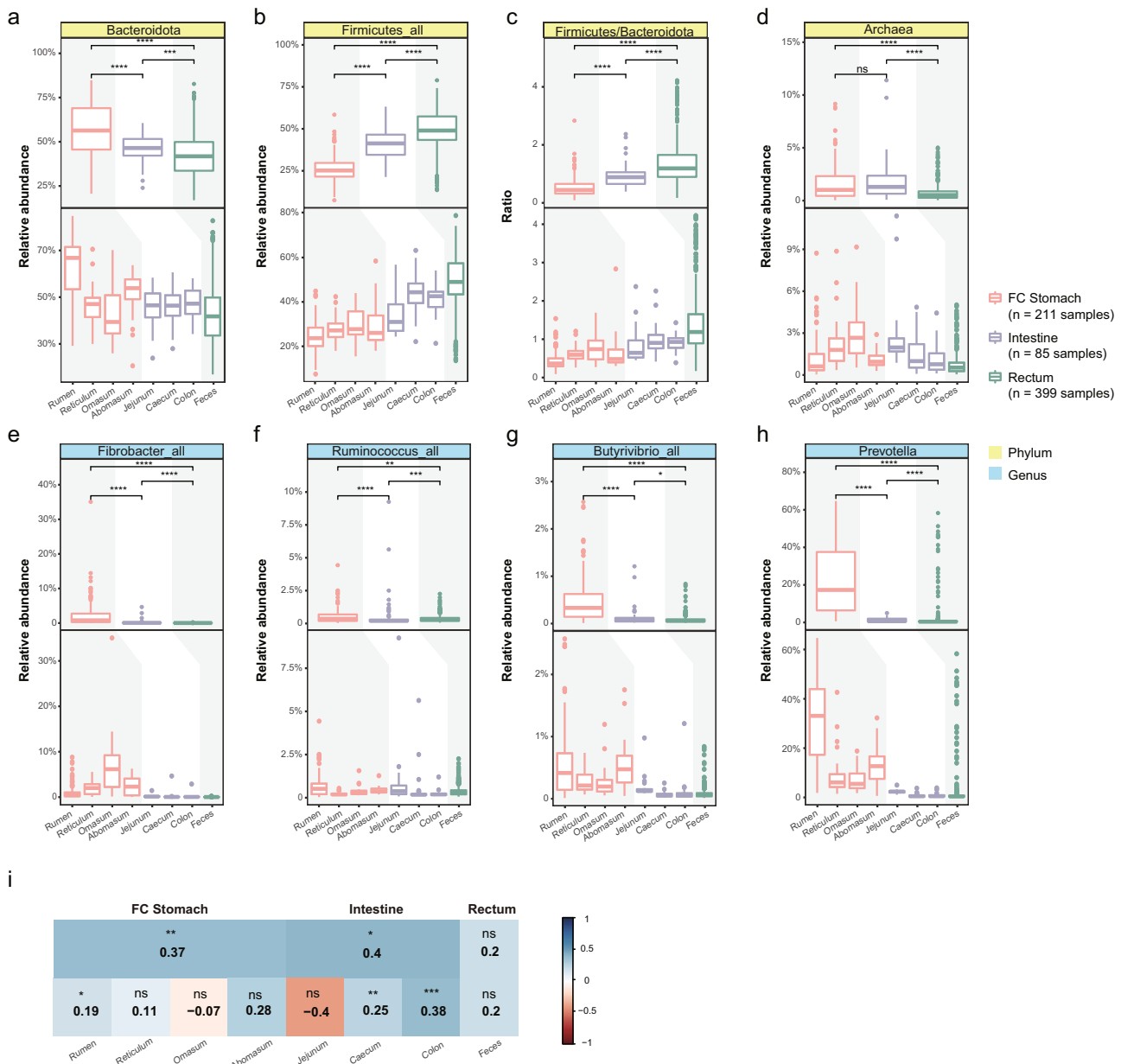

**Fig. 4 Distinct abundance patterns of MAGs correlate with the functions of the digestive tract.** In panels **a**–**h** (except **c**), boxplots are used to summarize the abundance distributions of the abundance of taxa of interest in the three sections (upper part) and eight sites (lower part) sampled; the *Y*-axis in these panels (except **c**) indicates the relative abundances, while that in **c** indicates the abundance ratios between Firmicutes and Bacteroidota. The background colours of the panel labels indicate the classification levels of microbes, while the different colours of the boxes indicate the different sections of the DT: FC stomach (orange, *n* = 211 samples), intestine (purple, *n* = 85 samples), and rectum (green, *n* = 399 samples). The pairwise Wilcoxon rank-sum test was used to compare the groups (the upper part). Boxplots show the median and the 25th and 75th percentiles, the solid lines indicate the minima and maxima, and the points falling outside of the whiskers of the boxplots represent the outliers. Level of significance: ns $P \geq 0.05$, *$P < 0.05$, **$P < 0.01$, ***$P < 0.001$, ****$P < 0.0001$. **i** Heatmap showing the Spearman correlation between archaea and *Fibrobacter_all* along the DT; the numbers are the correlation coefficients between archaea and *Fibrobacter_all*. The correlations of all phyla and genera from **a**–**h** are shown in Supplementary Fig. 9. The pairwise Wilcoxon rank-sum test was used to compare the groups. Level of significance: ns $P \geq 0.05$, *$P < 0.05$, **$P < 0.01$, ***$P < 0.001$, ****$P < 0.0001$.

abundances in the FC stomach, intestine, and rectum, respectively (Supplementary Data 4; Supplementary Fig. 10). In addition to those mentioned above, the other differentially abundant taxa may also contribute to the physiological functions of the corresponding DT sections and should be investigated further.

**Functional characteristics of MAGs in different sections along the digestive tract.** We next explored the proteomic contents of the buffalo metagenome and their putative functions. We predicted in total 9,470,238 proteins based on 4960 MAGs; after clustering by CD-HIT[91], we obtained a non-redundant proteome dataset of 5,862,748 proteins. We annotated these proteins by comparing the amino acid sequences with the eggNOG database[92] using eggNOG-mapper[93] and the CAZy[94] database using dbCAN2[95]. A total of, 4,787,680 proteins (81.7% out of total) could be annotated according to one or both methods; among which 114,989 and 4,672,691 were annotated by using CAZy and eggNOG, respectively.

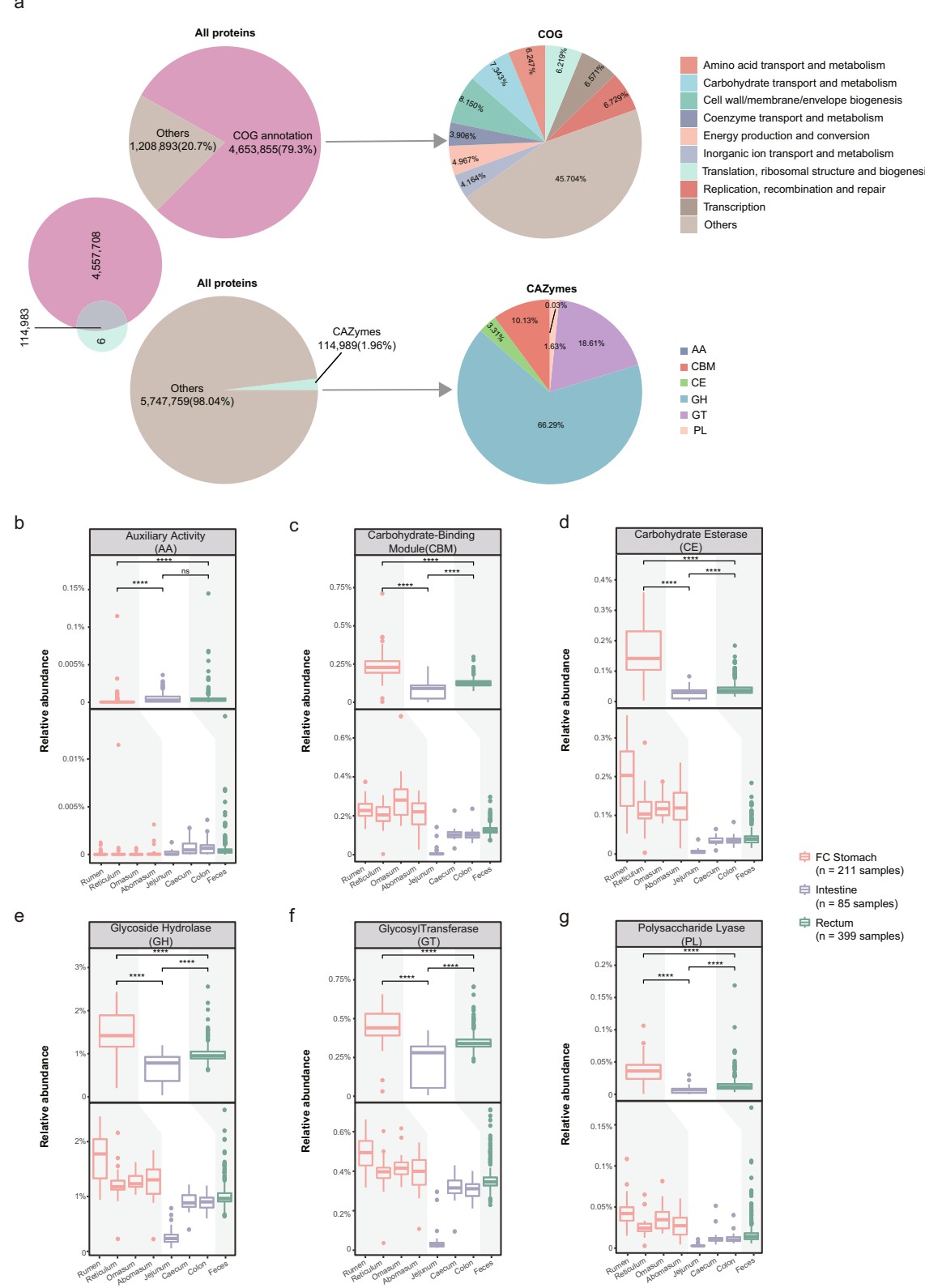

Only a small proportion of the proteins (114,989; 1.96% out of total) were annotated as enzymes according to CAZy (Fig. 5a). The majority of these protenins were annotated as glycoside hydrolases (GH, 76,224, 66.29% out of 114,989), after which glycosyltransferases (GT, 21,402, 18.61%) and carbohydrate-binding modules (CBM, 11,645, 10.13%) were the most annotated groups; the remaining three categories including carbohydrate esterases (CE, 3804, 3.31%), polysaccharide lyases (PL, 1879, 1.63%) and predicted auxiliary activities (AA, 35; 0.03%), contained far fewer proteins. Overall, the annotated proteins showed decent sequence identities against the proteins in the CAZy database, with an average identity of 59.11% (Supplementary Fig. 11); however, only a small proportion (3.8% out of the total annotated proteins) showed identities higher than 90%,

**Fig. 5 Functional annotation of the 5,789,172 non-redundant proteins encoded by buffalo MAGs and the distribution of their abundance along the DT.**
**a** Functional annotations of buffalo microbial proteins. Annotation results obtained using EggNOG-mapper (upper) and dbCAN2 (lower); pies show the proportions of proteins annotated by these two methods (left) and the overall categories (right). Boxplots summarize the abundance distributions of proteins (in functional groups) annotated by dbCAN2 including **b** auxiliary activities (AA), **c** carbohydrate-binding module (CBM), **d** carbohydrate esterase (CE), **e** glycoside hydrolase (GH), **f** glycosyltransferase (GT), and **g** polysaccharide lyase (PL) in the three sections (upper part) and eight sites (lower part). $Y$-axis: relative abundances (i.e. the sum of all proteins in a functional category). The different colours of boxes indicate the different sections of the DT: FC stomach (orange, $n = 211$ samples), intestine (purple, $n = 85$ samples) and rectum (green, $n = 399$ samples). The pairwise Wilcoxon rank-sum test was used to compare between the groups (the upper part). Boxplots show the median, and the 25th and 75th percentiles, the solid lines indicate the minima and maxima, and the points falling outside of whiskers of the boxplots represent the outliers. Level of significance: ns $P \geq 0.05$, *$P < 0.05$, **$P < 0.01$, ***$P < 0.001$, ****$P < 0.0001$. More details about the definition of the relative abundance of a protein can be found in the methods.

indicating a poor representation of our MAGs in public databases.

We then calculated the relative abundance of all proteins in each sample and compared the distributions of their abundance (see Methods). As shown in Fig. 5b–g, all six abovementioned CAZy families showed significant differences in terms of their relative abundances (the sum of the abundance of all proteins was 100%) among the three DT sections. Interestingly, all CAZy protein families except for the auxiliary activity (AA) family showed the highest abundances in the FC stomach, followed by the rectum and then the intestine, supporting the central role of the FC stomach in food digestion and processing (Fig. 5b–g); the rumen often showed the highest abundances of these CAZy families among all DT sites (except for the carbohydrate-binding module (CBM), for which the omasum displayed the highest abundance), although other FC stomach sites also contained significantly more abundant CAZy proteins than the other DT sections (Fig. 5b–g). Again, we identified the jejunum as an outlier site that contained the lowest percentages of all six CAZy families (Fig. 5b–g), likely due to its low microbial diversity (Supplementary Fig. 6).

**Comparisons of rumen microbiota between buffalo and cattle.**
The rumen is often is considered the most important FC stomach site in ruminants. The rumen is the first section of the DT and the largest of the FC stomach sites; it governs the first steps of feedstuff degradation, and is the main site of methane production[26]. The microbiota of the rumen plays important role in its functioning and was the first microbiota of the FC stomach to be studied by researchers[26,96]. We thus compared the taxonomic and functional profiles of the buffalo rumen microbiota with those of cattle which was recently made available by *Stewart* et al.[30]. To perform a fair comparison, we obtained 4941 rumen uncultured genomes (RUGs, similar to MAGs; see ref. [30].), encoding 4,879,163 non-redundant proteins and calculated their abundances in each sample using the same methods employed in this study (Methods).

As shown in Fig. 6a, we found that buffalo and cattle showed significant differences in the two most dominant phyla (Firmicutes_all and Bacteroidota) and consequently in the F/B ratios (Fig. 6a). Bacteroidota species, especially those in the dominant genus of the phylum, *Prevotella*, can utilize lactate[84,97,98] and are capable of degrading noncellulose plant fibres. Therefore, the higher abundances of Bacteroidota, *Prevotella*, and *Butyrivibrio_all* identified in the buffalo rumen than in that of cattle, as well as the similar levels of *Fibrobacter_all* (responsible for cellulolytic plant fibre digestion) between the two (Fig. 6a), suggested that buffalo is better adapted to coarse forage than cattle[88]. Conversely, we found significantly higher levels of Archaea and *Ruminococcus_all* in cattle, and these species all played important roles in methanogenesis through biohydrogenation and glycolysis (Fig. 6b), suggesting that buffalo may produce significantly less methane than cattle.

We also compared the abundance of rumen protein families (i.e. CAZy) between buffalo and cattle. Surprisingly, all six families were significantly abundant in buffalo (Fig. 6c). This result further suggested that buffalo has a greater carbohydrate metabolism capacity than cattle. In attention, comparison based on all CAZy proteins (Supplementary Fig. 12) revealed a significant abundance of GT and CBM in cattle, indicating that the rumen of cattle contained a higher proportion of enzymes related to the formation of glycosidic bonds than the rumen of buffalo.

We note that the differences observed here were likely due to the different diets of buffalo and cattle. However, due to limited dietary information available from the public dataset of cattle, we were unable to disentangle the effects of the diets on the differences in the microbial composition forom those of other factors. Future studies are thus needed for further elucidation of this issue.

**Discussion**
As an important livestock species, buffalo provides humans with milk, meat, leather, and draft power. Similar to other ruminants, the DT is the key to the quality and wellbeing of buffalo and shows intense interactions with microbes. However, the lack of microbial reference genomes from different sites of the DT has greatly hindered our understanding of the functional interactions between DT sites and their microbial ecology and our ability to modulate the physiology and economically important phenotypes of buffalo via the DT microbiota. More importantly, the comprehensive profiling of the methanogenic microbes that are presumably inhabited the rumen may provide us with insights into contribution to reducing the emission of methane, an important source of greenhouse gases. Recent studies on ruminant microbiota have mostly focused on the rumen, while explorations of the digestive tracts are still lacking. To fill in these gaps, we performed a comprehensive survey on the microbial ecology of the buffalo DT. We collected 695 samples from eight DT sites within three sections, including the FC stomach, intestine, and rectum. To further increase the representativeness of our study, we obtained samples from six different geographical (Guangxi, Henan, Anhui, Yunnan, Hainan, and Hubei; Supplementary Data 1, Supplementary Table 1; Supplementary Fig. 1a), three breeds (river, swamp, and hybrid buffalo; Supplementary Data 1, Supplementary Table 2), both sexes and two developmental stages (Supplementary Data 1, Supplementary Table 3).

We performed mNGS on these samples and obtained 4960 MAGs. These MAGs greatly improved the coverage of the raw sequencing reads from 64% in the public databases (the combination of reference genome datasets including BFAP genomes from NCBI RefSeq) to 85%. Taxonomic annotation revealed that all MAGs could be classified into known phyla; however, more than 90% of the MAGs were novel at the species level. Thus, our dataset represents a great expansion of the available buffalo microbiomes.

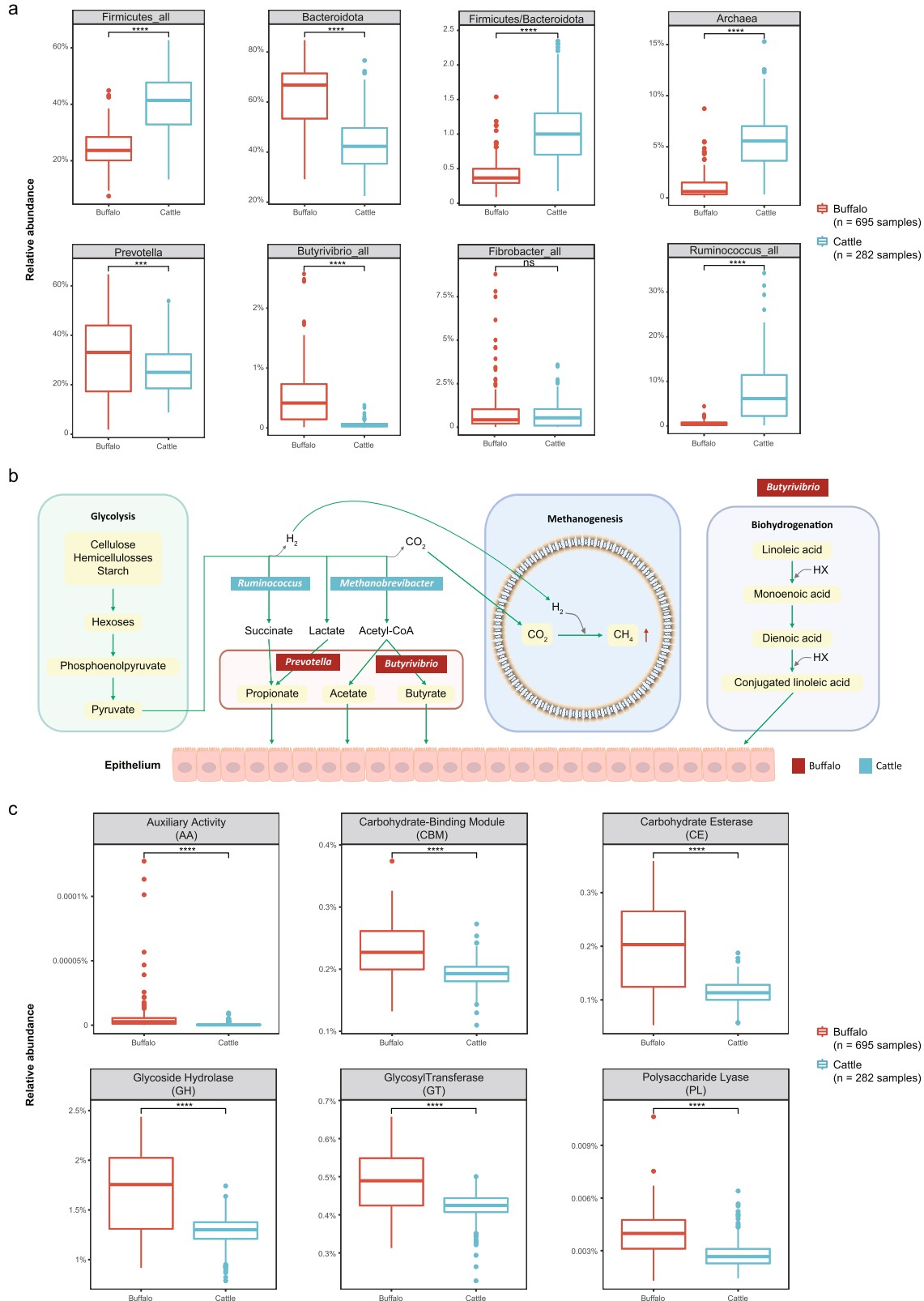

The samplings of different sites of the DT allowed us to better understand the functional associations between the DT sites and their microbial ecology. For example, we found that Firmicutes and Bacteroidota, the two most dominant phyla, showed distinct abundance patterns along with the DT: Firmicutes abundance increased along the digestive tract (Fig. 4a), while Bacteroidota showed the opposite pattern (Fig. 4b). Consequently, the Firmicutes to Bacteroidota ratio (F/B ratio, Fig. 4c) was lowest in the FC stomach and highest in the rectum. The F/B ratio is related to energy harvesting;[77,78] its trend thus coincides with the physiological transition from food digestion (FC stomach) to energy harvesting (intestine) along with the DT. Our data also allowed us to validate known interactions between microbes and DT sites, including the enrichment of fibre-digesting and

**Fig. 6 Comparisons of rumen microbiota between buffalo and cattle. a** Relative abundance of selected individual taxa between buffalo (red, $n = 695$ samples) and cattle (blue, $n = 282$ samples), except in panel 3 of the first row, which shows Firmicutes/Bacteroidota ratios. The pairwise Wilcoxon rank-sum test was used to compare between the groups. Boxplots show the median and the 25th and 75th percentiles, the solid lines indicate the minima and maxima, and the points falling outside of the whiskers of the boxplots represent the outliers. Level of significance: ns $P \geq 0.05$, *$P < 0.05$, **$P < 0.01$, ***$P < 0.001$, ****$P < 0.0001$. **b** Schematic diagram of plant fibre digestion and methane metabolism in ruminants. The key microbial genera in these processes that showed significant differences between buffalo and cattle are highlighted: red, significantly higher in the buffalo rumen; blue, significantly higher in the cattle rumen. **c** Comparisons of protein families between buffalo (red, $n = 695$ samples) and cattle (blue, $n = 282$ samples). The *Y*-axis shows the relative abundance of protein families; here, the relative abundance of a protein family is defined as the percentage of reads mapped to the coding sequences of member proteins in a family out of all reads mapped to all coding sequences (the sum of the abundances of all proteins is 100%). The pairwise Wilcoxon rank-sum test was used to compare between the groups. Boxplots show the median and the 25th and 75th percentiles, the solid line indicate the minima and maxima, and the points falling outside of the whiskers of the boxplots represent the outliers. Level of significance: ns $P \geq 0.05$, *$P < 0.05$, **$P < 0.01$, ***$P < 0.001$, ****$P < 0.0001$. GH glycoside hydrolase, GT glycosyltransferase, PL polysaccharide lyase, CE carbohydrate esterase, AA auxiliary activities, CBM carbohydrate-binding module.

methane-producing microbes in the FC stomach. Surprisingly, we found that *Fibrobacter*, a group of cellulolytic bacteria reported to mainly colonize the rumen, showed higher relative abundance in the omasum; its distributions coincided with that of archaea, the main methane producers, whose abundances also peaked in the omasum. Archaea were also highly abundant in both the FC stomach and intestine and showed positive correlations with *Fibrobacter*, indicating their roles in methane production in both sections. Based on the abundance of the microbes at the phylum and genus levels, we identified several features that are abundant in the FC stomach, intestine, and rectum. These results highlighted the importance of obtaining samples from all DT sites, especially sites other than the rumen.

We also evaluated the functional capacity of the microbial ecology of the MAGs at the DT sites by annotating the protein-coding genes of the MAGs and comparing them against the CAZy and eggNOG databases. We found that all six CAZy families showed significant differences among the examined DT sections, suggesting that they play different roles associated with distinct sections of the DT. Our data showed poor sequence identity with public data, indicating the existence of previously unidentified protein sequences and, thus, novel functions encoded by the DT-associated MAGs.

The rumen is the most important section in the DT; its microbiota has recently been extensively explored in cattle. We thus took the opportunity to compare the rumen microbes between these two closely related model ruminants. Our results showed significant differences between buffalo and cattle. For example, we observed a higher abundance of microbes with the ability to degrade fibres in the buffalo rumen than in that of cattle. In addition, the relative abundance of methane-producing archaeal species in buffalo was significantly lower than that in cattle, indicating less production of methane and more complete use of energy from feedstuff.

The microbial composition is affected by both intrinsic factors, such as the DT section, sex, breeds, and developmental stages, and environmental factors, such as geographical locations. In this study, we found that all examined factors exerted significant effects on the DT microbiota of buffalo in both single- and multifactor analyses (see Materials and Methods); notably, the DT section exerted the strongest effects, followed by geography, breed, developmental stage, and sex. Since the sampling strategy applied in this study was optimized to compare the microbial compositions across DT sections, we focused on the comparative analysis between DT sections and left the comparisons of other factors for future studies.

Our catalogue of microbial genomes and their encoded proteins represents the largest effort thus far to characterize the microbial ecology of all major sections of the DT. Our study provides insight into the microbial functions and interactions

with distinct DT sites, and has generated valuable resources for the community interested in microbial interventions for achieving better buffalo quality.

## Methods

**Sample collection.** A total of 695 samples were collected for metagenome sequencing. To ensure the diversity of the samples, they were collected from three breeds (153, 430, and 112 samples from river, swamp, and hybrid buffaloes, respectively), six regions (Guangxi, Henan, Anhui, Yunnan, Hainan, and Hubei in China), two sexes (females and males, respectively) and two developmental stages (adult buffaloes and calves; see Supplementary Data 1 for details). Among the 695 samples, 497 were obtained from live buffalo, including 98 rumen samples collected via a stomach tube and 399 faecal samples collected from the rectum by using disposable soft, long-arm gloves. The remaining 198 samples were all collected from slaughtered buffalo, including 31 rumen samples, 28 reticulum samples, 32 omasum samples, 22 abomasum samples, 24 jejunum samples, 31 caecum samples, and 30 colon samples (Supplementary Data 1, Supplementary Table 4). The eight sampled DT sites were divided into three sections: the FC stomach (rumen, reticulum, omasum, and abomasum), intestine (jejunum, caecum, and colon), and rectum (see also Fig. 1). All samples were immediately frozen after collection in liquid nitrogen and stored at −80 °C until DNA extraction. Most samples were subjected to DNA extraction within a week after collection.

**DNA extraction, library construction, and metagenomics sequencing.** Three grams of each sample was employed for DNA extraction. DNA was extracted via a bead-beating method using a mini-bead beater (Biospec Products; Bartlesville, USA), followed by phenol-chloroform extraction. The solution was precipitated with ethanol, and the pellets were suspended in 50 μL of Tris-EDTA buffer. DNA was quantified using a NanoPhotometer® (IMPLEN, CA, USA) following staining using a Qubit® 2.0 Fluorometer (Life Technologies, CA, USA). DNA samples were stored at −80 °C until further processing.

Library preparation was performed according to the TruSeq DNA Sample Preparation Guide (Illumina, 15026486 Rev. C) method and procedure using 500 ng DNA as the template. Qualified libraries were selected for paired-ended sequencing on the Illumina NovaSeq 6000 platform with a read-length of 150 base pairs (PE150).

**Quality control and removal of host- and food-associated genomes.** Paired-end raw sequencing reads were first trimmed by using Trimmomatic[99] (v.0.35) to remove vectors and low-quality bases. Sequences longer than 110 bases with an average base quality greater than 30 after trimming were retained for further analysis. To remove possible reads from the host or food, genomic sequences of buffalo[100], *Glycine max*[101], *Zea Mays*[102], and *Medicago truncatula*[103] were downloaded from NCBI and used as references in Bowtie2[104] (v.2.3.3) analysis with the options '-p 10 --very-sensitive'. Reads that aligned concordantly to references were removed as contamination. Thus 20.3% of the bases were removed on average (Supplementary Data 5, 6). The remaining "clean reads" were used for further analysis.

**Generation and quality assessment of metagenomics-assembled genomes (MAGs).** MEGAHIT (v.1.2.8) and metaSPAdes (v.3.13.0) were used for single-sample assembly. MEGAHIT automatically determines the best k-mer value; however, metaSPAdes allows users to manually select a range of k-mers. To test the best k-mer parameter, we randomly selected 16 samples and performed their assembly using metaSPAdes with seven different k-mers ranging from 21 to 141 with a step size of 20. We then used two quality metrics, N50 and assembled reads, to test the assembly results. For each of the tested samples, we selected two sets of resulting scaffolds, one of which showed the best N50 (referred to as "Kmer_N50), while the other showed the greatest number of assembled reads (Kmer_MRate).

Because modern assemblers including metaSPAdes, do not report the number of assembled reads, we used BWA-MEM[105] (v.0.7.15) to align the reads to the scaffolds and calculated the mapping rate. As shown in Figure S11, we found significantly higher proportion of assembled reads (mapping rates) in the Kmer_MRate group than in the Kmer_N50 group; this was expected because the Kmer_MRate group was optimized for higher mapping rates. For comparison, we also used MetaBAT2[65] (2.12.1) to group the scaffolds into bins using the generated scaffolds for each sample and calculated the mapped rate of the reads to the bins. We did not find a significant difference in the mapping rates between the two groups (Supplementary Fig. 13 and Supplementary Data 7). Although the Kmer_MRate method assembled more reads than the Kmer_N50 method at the scaffold level, the two did not show significant differences at the scaffold level. Furthermore, we tested the effects of the quality metrics (Kmer_MRate vs. Kmer_N50) on the assembled bins. All bins were aggregated and dereplicated using dRep[66] (v.2.3.2), followed by CheckM (v.1.0.18) for quality assessment. When the -sa = 0.99 option of dRep was used (i.e. the MAGs were dereplicated at the strain level), we obtained more MAGs using the Kmer_MRate method; however, the Kmer_MRate method generated significantly fewer high-quality MAGs (69 vs. 85 under the Kmer_N50 method; Supplementary Fig. 14a). Additionally, the MAGs generated by Kmer_MRate presented significantly shorter N50s (Supplementary Fig. 14B) and genome lengths (Supplementary Fig. 14C), and were significantly fragmented (Supplementary Fig. 14D). We found similar results at the species level (i.e. MAGs dereplicated using –sa = 0.95). In addition, the Kmer_MRate required an additional 60 CPU hours per sample per k-mer on average to align the reads to the scaffolds; thus, we thus used the Kmer_N50 method to select the best k-mer in this study.

Then, for each sample, we compared the total length, N50, and contig numbers of the assembly results from the two assemblers, and selected the assembly with a relatively long total length and N50 as the final assembly of this sample. Coassembly was performed for each of the three sections by combining all samples as the input using only MEGAHIT because of its lower time and memory consumption relative to metaSPAdes. However, as shown in Supplementary Fig. 15, metaSPAdes could generate contigs with a longer N50 than MEGAHIT through comparison with N50 values among other measurements of assembly quality. Thus, the assembly results of the two tools were used to increase the coverage and quality of the resulting contigs (single-sample results were binned, and the resulting bins were combined).

MetaBAT2[65] (2.12.1) was used to group contigs into bins. First, BWA-MEM[105] (v.0.7.15) was used to map reads to the contigs (MEGAHIT) and scaffolds (metaSPAdes) to obtain the depths of the contigs (MEGAHIT) or scaffolds (metaSPAdes) in each sample. The results were saved in SAM files. Then, SAMtools[106] (v.1.8) was used to convert the SAM files to BAM format. Finally, MetaBAT2 was used to calculate coverage from the resulting BAM files and output the results of the bins. As a result, single-sample binning produced a total of 58,041 bins, while an additional 53 bins were obtained from coassembly binning. We referred to these bins as MAGs.

All bins were dereplicated using dRep[66] (v.2.3.2) with the option 'dereplicate_wf -p 16 -comp 80 -con 10 -str 100 -strW 0 –nc 0.1'. The '-nc 0.1'parameter is recommended by the authors of the dRep paper. However, we noted that different options were used in these publications(e.g. '-nc 0.3' in Almeida et al.[107] and '-nc 0.6' in Almeida et al.[70]). We thus tested four different options of the '-nc' parameter (i.e. -nc = 0.1, 0.2, 0.3, or 0.6) in selected samples at the strain (-sa 0.99) and species (-sa 0.95) levels. We selected four groups of samples to test the '-nc' parameter. For each group, we obtained the same number of dereplicated bins regardless of the '-nc' options (Supplementary Data 8). We also combined the bins from all samples and performed dereplication, and we again obtained the same number of dereplicated bins (Supplementary Data 9). Therefore, the default threshold (0.1) of the "-nc" option of dRep was used in this study. After binning, CheckM[67] (v.1.0.18) was used to assess the quality of the resulting MAGs. After removing MAGs with < 80% completeness or > 10% contamination, the remaining MAGs were processed via two clustering steps to remove replicates with the default parameters. The first step involved a rapid primary algorithm (Mash, ANI = 0.9), and the second involved a more sensitive algorithm (ANI = 0.99 and 0.95). Genomes with <99% ANI belong to different strains, while those with <95% ANI belong to different species. Then, we removed MAGs larger than 10 Mb. Finally, 4960 non-redundant strain-level and 3255 species-level MAGs were obtained.

To calculate the coverage of each MAG in each sample, the clean reads of each sample were mapped to the 4960 MAGs using BWA-MEM with the default parameters. After converting the SAM files to BAM format using SAMtools[106] (v.1.8), BEDTools[74] (v.2.27.1) was employed to calculate the coverage of the MAGs, which was defined as the total bases mapped to a MAG in a sample divided by its length.

**Comparisons with reference microbial genomes and MAGs associated with model organisms**. To determine whether our MAGs could improve the coverage of microbial genomes associated with the buffalo DT, MAGs of model organisms including humans[70], chickens[57], pigs[53], and cattle[30], were downloaded from their respective sources. In addition, a BFAP dataset including reference microbial genomes (bacterial, fungal, archaeal, and protozoan) was generated from the NCBI RefSeq genome database and the Hungate collection genomes[71].

BWA-MEM was used to map the "clean reads" to the above datasets and our MAGs as references. The mapping rate was calculated for each sample as the percentage of clean reads mapped to each of the reference datasets.

To compare the sizes of the buffalo MAGs with public datasets, we also downloaded the NCBI RefSeq prokaryotic genomes with assembly levels of 'chromosome' and 'complete' assembly levels; as of June 28, 2021, this dataset consisted of 457 archaeal and 28,011 bacterial genomes.

**Taxonomic assignments of buffalo MAGs**. The taxonomic assignment of the MAGs was performed using the GTDB-TK (v.1.4.1)[73] via its "classify_wf" workflow. The results were visualized in GraPhlAn (v.1.1.3))[108] as a phylogenetic tree.

**Annotation and functional characterization of MAG-encoded proteins**. MAG-encoded proteins were predicted using Prodigal[109] (v.2.6.3), tRNA genes were annotated using tRNAscan-SE (v 2.0) and 16 S rRNA genes were predicted using Barrnap (v 0.9). CD-HIT[91] (v.4.8.1) was used to cluster predicted proteins with the option '-c 0.95 -n 10 -d 0 -M 16000 - T 8'. The resulting non-redundant proteins were searched against the CAZy database using dbCAN2[95] and the EggNOG database using eggnog-mapper (v.4.5)[92].

**Estimation of the relative abundance of MAGs and proteins in each sample**. To enable the identification of differentially abundant MAGs and proteins in different sites/sections along with the DT, the relative abundance of MAGs and proteins was calculated based on their respective coverage in each sample. The coverage of each protein was calculated via the same method described aboce for the coverage of MAGs, and the relative abundance of a MAG (gene) in each sample was calculated as the percentage of the coverage of the MAGs (protein) relative to the sum of the coverage values of all MAGs (proteins).

**Effects of host and environmental factors on the microbiota compositions**. PERMANOVA implemented in the R package 'vegan' was used to investigate the effects of host and environmental factors, such as geography, DT sections, developmental stages, breed, and sex, on the microbiota composition. PERMANOVA is a permutation-based extension of multivariate analysis of variance to a matrix of pairwise distances that partitions within-group and between-group distances to permit the assessment of the effect of an exposure or intervention (grouping factor) on the sampled microbiome[110]. We performed both single- (i.e. consider one factor at a time) and multifactor (i.e. considering all factors at the same time) analyses.

**Identification of differential taxa between groups**. To identify microbes that showed significant differences in abundances between sample groups of interest, we first calculated the relative redundancy of the MAGs in all samples. We mapped the clean reads of a sample to all MAGs and calculated the relative abundance of a MAG as the percentage of reads mapped to the MAG among the total reads mapped to all MAGs in the sample; thus, the total relative abundance of all MAGs in a sample is always 100%. Relative abundances at higher taxonomic levels, such as genera, families, and orders, were also determined by summing the abundances of their daughter clades according to the phylogenetic tree provided by GTDB-TK.

LDA implemented in the LEfSe tool[90] was used to identify differential taxa between groups of samples. We chose those with LDA scores >2 as differential taxa. The Wilcoxon test was used to reveal the statistical significance of the relative abundances of the differential taxa between groups. Then, we selected taxa that are known to play an important role in physiological functions related to the grouping context.

**Identification of genera positively related to archaea**. To identify the genera that were positively related to archaea, we calculated the correlations between archaea and all other major taxa at the phylum and genus levels. The relative abundance values of all the genera of the archaea phylum were combined, and the genera that were positively related to the archaea phylum were selected based on the significance of the correlations among all the relevant results. Then, a search was performed to obtain information on the genera, and the genera, whose functional information could not be retrieved were manually removed.

**Ethics approval**. The investigation was approved by the Experimental Animal Ethics Committee, College of Animal Science and Technology, Guangxi University, under reference number Gxu-2021-010.

**Statistics**. All processed data, unless otherwise stated, were loaded into R (v.3.6.3, https://www.r-project.org/), for analyzed or visualization.

**Reporting summary**. Further information on research design is available in the Nature Research Reporting Summary linked to this article.

## Data availability

The raw sequencing data used in this study are available in the NCBI SRA database under accession code PRJNA656389. The 4960 strain-level MAG data used in this study are available in the Figshare database under accession code 17000302. The data of the 3255 species-level MAG data used in this study are available in the Figshare database under accession code 17081366. Source data are provided with this paper and these data have been made public. Source data are provided with this paper.

## Code availability

The software parameters and scripts used in the analysis have been uploaded on GitHub (https://github.com/fengtong-bio/MEABDT).

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

## Acknowledgements

Q.Y.L. acknowledges support from the National Natural Science Fund (U20A2051, 31760648, and 31860638), the Guangxi Natural Science Foundation (AB18221120), and the Guangxi Distinguished Scholars Program (201835). We acknowledge Prof. Jiaxiang Huang, Dr. Guangsheng Qin, and Hui Li for providing partial buffalo samples. We also acknowledge Prof. Jianghua Shang, Assistant Prof. Chunyan Yang, Dr. Jieping Huang, Dr. Hui Li, and Miss Siwen Wu for manuscript polishing and editorial assistance.

## Author contributions

Q.L., W.H.C., and G.H. designed and directed the research; F.T., Z. Li, K.C., Y.L., Z. Liu, Y.X., B.L., C.Y., L.Y., D.S., and Q.L. participated in the sampling and background investigation of buffalo; T.W., N.L.G., and F.T. analyzed the data and wrote the paper with results from all authors; S.W. and Y.D. participated in the production of the chart; A.P. participated in the revision of the article. All authors read and approved the final manuscript.

## Competing interests

The authors declare no competing interests.
