## [Peer Review File · Nature Communications]

REVIEWER COMMENTS

Reviewer #1 (Remarks to the Author):

The authors present a comprehensive dataset of buffalo digestive tract microbiome, collected from multiple sites along the gastrointestinal tract (GIT) over a range of six provinces and comprising three different breeds. Overall the authors sampled 695 samples with an average of 41 million reads per sample. After what appears to be a state of the art assembly process, the authors have produced 4960 MAGs with completeness of $\geq 80\%$ and contamination $\leq 10\%$. The authors begin by presenting their experimental setup and data quality, including benchmarking it compared to other datasets. They continue to explore their data for its taxonomic composition and annotation parameters using the GTDB-TK program, the overall alpha and beta diversity, and demonstrate sample by sample the taxonomic distribution in their dataset. The authors performed an analysis of phyla abundance and ratio across the GIT and attempted to link several genera to archaeal abundance. Next, they present an overview of the different microbial functions with an emphasis on carbohydrate degradation. Lastly, a comparison between the buffalo and cattle microbiome with a suggestion for a possible pathway for microbial activity is presented.

Although the authors present a comprehensive dataset with high quality and well balanced in terms of host distribution, In my view, the current work lacks the scientific merit for Nature Communications. While the authors show extremely high bioinformatics capabilities and, as aforementioned, the dataset presented here is comprehensive with high quality and importance to the field, the overall analyses performed on the given dataset are overly shallow and lack any scientific merit. The authors did not offer any scientific question or hypothesis regarding their research, a mere presentation of the data is not enough, and most of the scientific conclusions presented in this study were published in previous studies such as DOI: 10.1038/srep14567 and DOI: 10.1038/srep42189. I want to commend the authors for the graphical representation in this study which is very good, clear, and simple, the study design is sound, but without a novel question, this is simply not enough.

Comments:

Study design - The authors should clarify how the samples were extracted from each GIT compartment. Were the animals culled prior to sample collection or subjected to a medical

procedure such as colonoscopy? The authors did not display any ethical approval for their study and should address this notion also.

In figure 1F, the authors show that some MAGs are less than 2Mb in length (even 0.5M). Considering their claim that the completeness percentage was above 80%, can the authors elaborate about these relatively short genomes? What microbial species do they represent? Can they compare their length to similar microbes? They seem too short for microbial genomes.

In figure 1H, The authors compared the coverage of raw metagenomic reads to the MAGs of both their datasets and other reference microbial MAGs originating from other model organisms. First, the authors need to clarify the biological relevance of this analysis. Although the authors state in line 175 that they wanted to check whether their MAGs could improve the coverage of microbial genomes associated with buffalo's DT, it is not understood how this analysis facilitated achieving this goal. Furthermore, what is the purpose of comparing other hosts, and what is the biological question and conclusion behind this comparison? If you claim that you see increased coverage for buffalo's DT, you should compare your reads to other buffalo datasets such as Zhang et al. (2017), cited by the authors earlier. If this analysis aimed at examining the specificity of their dataset, it could be biased because the assembled reads will always have higher specificity to the bins assembled from within these reads compared to the reads of other animals which were not part of the assembly process. I also urge the authors to perform the opposite analysis, i.e., mapping the raw reads from other datasets to their MAGs. Last, I must comment that your reference for rumen metagenomes (37) is wrong. Please correct your references.

Regarding the analysis shown in figure 2A, can the authors explain why the Firmicutes split into three clades? What's the source of the split? Could this split stem from the annotation tool (GTDB)?

Figure 2A has no merit in the field of rumen and gut microbiome. I understand the importance of introducing your data, but this should not be displayed as the main figure and better suited for the supplementary data.

The term "stomach", which is widely used by the authors, is not accurate. The reticulorumen is not a stomach. The actual "stomach" is only the abomasum. "stomach" should be replaced by the "upper digestive" tract or "foregut".

Figure 3 begins with an excellent representation of MAGs distribution across the GIT. The alpha diversity measurement analysis displayed afterward is not innovative enough to be in the central figure. I understand the necessity to show the intra-sample diversity in each compartment and sub-compartment, but this analysis has no innovation. The authors did not try to "develop" this analysis into any downstream biological questions that may arise from their findings. Later on, the authors

performed the requested beta diversity analysis. Despite its importance, this analysis is not innovative and has been widely presented in many GIT studies in multiple different hosts. Furthermore, can the authors relate to the horseshoe effect here? It is important to look at geographical locations that might explain the sparse clustering seen here, age as well.

The beautifully displayed per taxa sample analysis performed in figure 3E shows evident variation within a single body part. Why does it exist? Could other elements like geography, age, and diet have an effect here? As previously mentioned, the authors did not perform any downstream analysis and did not develop the biological question.

The analysis presented in figure 4 shows no innovation and is widely performed. The authors did not elaborate on how they decided to focus on the four prokaryotes (Fig.4 e-h). Why did the authors determine that these are important to look at? Was it based on previous knowledge? I see no statistical test showing if the selected species have a higher correlation with archaea than other randomly chosen species. Other species also have the functions related upstream functions to methane production. I am missing the biological explanation for the choice. Furthermore, a quick web search will reveal that such analysis has been performed before.

Functional annotation analysis: Please elaborate more on your ORF\protein prediction process. I did not understand whether the COG was done in a supervised manner. Furthermore, was your process blind to sequences not framed by an ORF? I want to mention again that the analysis itself has been performed on buffalos in a previous study cited by the authors (Zhang et al. 2017). The authors did not seem to extend their analysis beyond the scope of the introduction.

Comparing their dataset to cattle data is problematic as cattle themselves are very varied in dietary composition, purpose (dairy versus meat cattle), and breed. Did your analysis correct for these variates? Please be more specific regarding the nature of this comparison.

Last analysis- The authors should notice that this comparison is rather generalistic and that the abundance of these bacteria could vary in different cattle breeds. I am not sure that the prevotella utilize lactate. Please check and correct accordingly if needed.

Reviewer #2 (Remarks to the Author):

The authors present an interesting and comprehensive analysis of the buffalo digestive tract from a metagenomics perspective

Specific comments:

line 89 misses some references:

Cow/cattle:

<https://www.nature.com/articles/s41587-019-0202-3>

<https://www.nature.com/articles/s41598-021-81668-9>

<https://genomebiology.biomedcentral.com/articles/10.1186/s13059-020-02144-7>

<https://www.biorxiv.org/content/10.1101/2021.04.02.438222v1>

Reference 36 says sheep but the paper is from goat

Chickens:

<https://genomebiology.biomedcentral.com/articles/10.1186/s13059-020-1947-1>

<https://peerj.com/articles/10941/>

line 116 - were the reads paired-end?

Lines 157 and 159 have different reference formats? []

Line 159 references cattle MAGs and reference 37, but 37 is not a valid reference here

Line 161 and 162, high quality MAGS as defined by Bowers et al include full 16S gene and most tRNA genes - are these cut-offs included in the analysis here?

Line 420 and elsewhere - "better" is not the correct word here! I suspect highly adapted western breeds of cattle are highly efficient at carbohydrate degradation of certain carbohydrates; the differences with buffalo will almost certainly be due to differences in diet

505 - sample collection - how were they sampled? Stomach tube? After slaughter? Some other way?

Also how long were they stored in the freezer?

lines 540 - 543 - this is an unusual way of using SPAdes which usually combines data from multiple kmers

549 - combining data from two different assemblers needs to be described more - how was this done?

Line 559 when using dRep, the authors should state and justify which overlap threshold they used (-nc)

Data availability - I cannot assess the public data as the authors have not made it available.

Index

Response to referee 1	1
Response to referee 2	17

Response to referee 1

NOTE: for convenience purposes, we broke the reviewers' comments into relevant sections, numbered them and provided point-to-point responses. We did not truncate nor rephrase the comments by any means.

General remark

The authors present a comprehensive dataset of buffalo digestive tract microbiome, collected from multiple sites along the gastrointestinal tract (GIT) over a range of six provinces and comprising three different breeds. Overall the authors sampled 695 samples with an average of 41 million reads per sample. After what appears to be a state of the art assembly process, the authors have produced 4960 MAGs with completeness of $\geq 80\%$ and contamination $\leq 10\%$. The authors begin by presenting their experimental setup and data quality, including benchmarking it compared to other datasets. They continue to explore their data for its taxonomic composition and annotation parameters using the GTDB-TK program, the overall alpha and beta diversity, and demonstrate sample by sample the taxonomic distribution in their dataset. The authors performed an analysis of phyla abundance and ratio across the GIT and attempted to link several genera to archaeal abundance. Next, they present an overview of the different microbial functions with an emphasis on carbohydrate degradation. Lastly, a comparison between the buffalo and cattle microbiome with a suggestion for a possible pathway for microbial activity is presented.

Although the authors present a comprehensive dataset with high quality and well balanced in terms of host distribution, In my view, the current work lacks the scientific merit for Nature Communications. While the authors show extremely high bioinformatics capabilities and, as aforementioned, the dataset presented here is comprehensive with high quality and importance to the field, the overall

analyses performed on the given dataset are overly shallow and lack any scientific merit. The authors did not offer any scientific question or hypothesis regarding their research, a mere presentation of the data is not enough, and most of the scientific conclusions presented in this study were published in previous studies such as DOI: 10.1038/srep14567 and DOI: 10.1038/srep42189. I want to commend the authors for the graphical representation in this study which is very good, clear, and simple, the study design is sound, but without a novel question, this is simply not enough.

Response:

Before we answer the reviewer's questions, we would like to thank the reviewer for his/her time and energy to help us improve this study.

Although we do agree with the reviewer that this study focused more on the data collection and presentation, we do have clear scientific questions as detailed below:

- First, we would like to know what microbes are associated with all sections of the buffalo's digestive tract (DT), especially those did not have genomic sequences in public databases. To overcome the shortcomings of previous studies that either focused only on a specific DT site or only had limited samples, we presented a comprehensive survey on the microbial ecology along buffalo's DT, by collected 695 samples from eight DT sites. To maximize the microbe discovery, we took samples from buffalos that varied in **geography, breed, developmental stage and sex**. Through mNGS sequencing we obtained 4,960 high-quality metagenome-assembled genomes (MAGs), representing ~85% of all raw reads. More than 90.7% of these MAGs are unclassified at species level. From the MAGs we annotated 5,862,748 non-redundant proteins, a significant proportion of which was not found in public databases. Thus, these datasets represented significant improvement on the coverage of buffalo-associated microbes.
- Second, we wished to identify microbes that showed differentially abundant microbes between DT sections and/or sites in an unbiased manner. Previous studies might have identified such associations between specific sites and previously known microbes, but a comprehensive survey was not yet available. In this study, we first confirmed and extended previously known associations between microbes and DT sites including the enrichment of fiber-digesting

and methane-producing microbes in the four-chambered stomach, and then revealed novel associations, including *Fibrobacter*, a group of cellulolytic bacteria known to colonize mainly in rumen, was found to have higher abundance in omasum. Its co-distribution with archaea suggested its roles in methane-production at both rumen and omasum. In addition, we also identified associations between DT sites/sections and previously uncharacterized microbes; the functional roles of the latter should be further explored.

With that said, we would also like to argue that comprehensive profiling is very important and can significantly promote the development of the related field, such as the draft human genome and other various genome sequencing projects, and the human gut gene catalog and many other gut genome catalogs that followed. Our data and results provided researchers with a unbiased view of the buffalo's MAGs, their encoded genes, and their associations with different DT sites; we believe these would lay the foundations for future research on the functions of the microbes and their roles in the different DT sites, and intervention for better buffalo's quality through DT microbes.

Comments for the Author

Comment #1

Study design - The authors should clarify how the samples were extracted from each GIT compartment. Were the animals culled prior to sample collection or subjected to a medical procedure such as colonoscopy? The authors did not display any ethical approval for their study and should address this notion also.

Response:

We thank the reviewer for the reminder, and apology for the omission.

Regarding the sampling methods, out of the total 695 samples, 497 were taken from live buffalo, including 98 rumen samples collected by stomach tube and 399 fecal samples collected by disposable soft long arm gloves from rectum. The remaining 198 samples were all from slaughtered buffalo, including 31 rumen samples, 28 reticulum samples, 32 omasum samples, 22 abomasum samples, 24 jejunum samples, 31 cecum samples and 30 colon samples.

Actions:

1. We added the above information to lines 538-543, now this part reads: "Out of the total 695 samples, 497 were taken from live buffalo, including 98 rumen

samples collected by stomach tube and 399 fecal samples collected by disposable soft long arm gloves from rectum. The remaining 198 samples were all from slaughtered buffalo, including 31 rumen samples, 28 reticulum samples, 32 omasum samples, 22 abomasum samples, 24 jejunum samples, 31 cecum samples and 30 colon samples (Supplementary Table 1, 8).”;

2. Ethical approval was included in the “Ethics declarations” section in the previous version. We uploaded a photocopy of the approval as a supplementary information (Ethics_declarations.pdf).

Comment #2

In figure 1F, the authors show that some MAGs are less than 2Mb in length (even 0.5M). Considering their claim that the completeness percentage was above 80%, can the authors elaborate about these relatively short genomes? What microbial species do they represent? Can they compare their length to similar microbes? They seem too short for microbial genomes.

Response:

The MAGs that are less than 2Mb in size belong to mostly Firmicutes_A and Bacteroidota at the phylum level, and are mostly unclassified at the species level; the lengths and taxonomic classifications of all MAGs can be found in Supplementary Table 2.

In this study CheckM was used to evaluate the qualities of the generated MAGs. CheckM is a state-of-art tool in the field that calculates the completeness and contamination of a genome/MAG by using marker genes that are specific to a genome's inferred lineage within a reference genome tree (<https://ecogenomics.github.io/CheckM/>).

As compared the complete genomes in the NCBI Refseq prokaryotic genomes (457 archaeal and 28,011 bacterial genomes with assembly levels of complete or chromosome; downloaded as of June 28, 2021), our MAGs are indeed shorter (see the new Supplementary Figure 2a). However, due to the lack of representativeness of buffalo microbial genomes in public databases, the differences could in part due to the distinct characteristics of buffalo's MAGs. In addition, we believe that the completeness as measured by the presence of key genes, is a better index than genome

sizes, since the latter are known to vary significantly even within species. For example, we also compared the completeness and genome size of buffalo's MAGs with the 4,941 cattle rumen MAGs from Stewart *et al*, we found that although buffalo MAGs are shorter than the cattle rumen MAGs (new Supplementary Figure 2b), the former had significantly higher complete measurements than the latter (new Supplementary Fig.2c). These results may suggest that buffalo-associated microbes tended to use compact genomes; however, further validation is needed for this conclusion.

Actions:

1. As the reviewer suggested, we downloaded NCBI Refseq prokaryotic genomes (457 archaeal and 28,011 bacterial genomes) with assembly levels of complete or chromosome, and compared their genome sizes with the buffalo MAGs. We added a brief summary on the methods to the Materials and Methods section at lines 615-617; now this part reads: "To compare the sizes of buffalo MAGs with public datasets, we also downloaded the NCBI Refseq prokaryotic genomes with assembly levels of chromosome and complete; as of June 28, 2021, this dataset consisted of 457 archaeal and 28,011 bacterial genomes.";
2. We summarized the results into a density plot, and added it as a new Supplementary Fig.2a;
3. We also compared the completeness values and genome sizes between buffalo and cattle rumen MAGs, and summarized the results into a new Supplementary Fig. 2b,c.
4. We then added a brief summary on the results to the revised manuscript at lines 158-170; now this part reads: "As compared with the complete genomes in the NCBI Refseq prokaryotic genomes (457 archaeal and 28,011 bacterial genomes with assembly levels of complete or chromosome; downloaded as of June 28, 2021), our MAGs are indeed shorter (Supplementary Fig.2a). However, due to the lack of representativeness of buffalo microbial genomes in public databases, the differences could in part due to the distinct characteristics of buffalo's MAGs. In addition, we believe that the completeness as measured by the presence of key genes, is a better index than genome sizes, since the latter are known to vary significantly even within species. For example, we also compared the completeness and genome size of

buffalo's MAGs with the 4,941 cattle rumen MAGs from Stewart *et al*, we found that although buffalo MAGs are also shorter than the cattle rumen MAGs (Supplementary Fig.2b), the former had significantly higher complete measurements than the latter (Supplementary Fig.2c). These results may suggest that buffalo-associated microbes tended to use compact genomes; however, further validation is needed for this conclusion.”.

Comment #3

In figure 1H, The authors compared the coverage of raw metagenomic reads to the MAGs of both their datasets and other reference microbial MAGs originating from other model organisms. First, the authors need to clarify the biological relevance of this analysis. Although the authors state in line 175 that they wanted to check whether their MAGs could improve the coverage of microbial genomes associated with buffalo's DT, it is not understood how this analysis facilitated achieving this goal. Furthermore, what is the purpose of comparing other hosts, and what is the biological question and conclusion behind this comparison? If you claim that you see increased coverage for buffalo's DT, you should compare your reads to other buffalo datasets such as Zhang et al. (2017), cited by the authors earlier. If this analysis aimed at examining the specificity of their dataset, it could be biased because the assembled reads will always have higher specificity to the bins assembled from.

Within these reads compared to the reads of other animals which were not part of the assembly process. I also urge the authors to perform the opposite analysis, i.e., mapping the raw reads from other datasets to their MAGs. Last, I must comment that your reference for rumen metagenomes (37) is wrong. Please correct your references.

Response:

We thank the reviewer for the comments. We will address the concerns in details below.

First, Figure 1H was generated for two reasons:

- The first one was to show that our MAGs could represent most of the sequencing reads generated in this study. A high mapping rate indicates that majority of the microbial genomes in a sample could be assembled into MAGs. With an average mapping rate of 84%, we considered that this mission was accomplished. Ideally, we should be able to find other metagenomic datasets for buffalo from public databases and show that our MAGs could also represent most of the public sequencing reads. But unfortunately, as of June 1, 2021, we did not find any mNGS datasets for buffalo in NCBI SRA, EBI ENA and other popular depositories for raw sequencing data. We did find the Zhang *et al* (2017) dataset mentioned by the reviewer, but it contained amplicon data (16s rRNA) from one sample. Therefore, we cannot directly compare our dataset to it, nor the other way round.
- The second reason was to check how well the microbial genomes from buffalo DT could be covered by public datasets such as MAGs from model organisms and Refseq prokaryotic genomes. The lower the coverage, the higher proportion of our data is novel (i.e., not found in public databases). This analysis is necessary because some buffalo associated microbes have been isolated and sequenced, so that their genomic sequences could be found in NCBI Refseq (BFAP); some microbes could have broad habitats or associate with other animals, so that they could be found in MAGs of other model organisms such as cattle, pig and human. Ideally this should be done at the genome level, i.e., to directly compare if our MAGs with those in the public datasets; in fact, we have done so using GTDB-Tk. However, due to different assembly strategies (e.g., tools and parameters) of these datasets and uncertainties in their qualities, these MAGs and prokaryotic genomes are not directly comparable with ours. Therefore, in this study we adopted a strategy that has been widely used in this field (such as PMIDs: 30661755, 31375809 and 32051016): we aligned the clean reads to selected public datasets and calculated the mapping rates. The latter represented the upper limit that the microbial genomes from buffalo DT could be covered by public datasets.

Thus, the high mapping rates to our MAGs indicate that the latter could represent most of the sequencing reads generated in this study, while the improved mapping rates

as compared with other datasets indicate the novelty of our dataset. Both analyses are necessary and widely used in this field.

Regarding the dataset by Zhang *et al* (2017), although the authors did both metagenomic and 16s rRNA sequencing for 10 buffalo fecal samples, only the amplicon data (16s rRNA) from one sample was available in public database (<https://www.ncbi.nlm.nih.gov/sra/SRP075434>). It thus could not be used for comparison with ours.

Mapping the raw reads from other datasets to our MAGs, as the reviewer suggested, is doable. For example, in the previous version of our manuscript, we compared the raw reads from the cattle rumen to our MAGs and found an average mapping rate of 71%. However, as we explained earlier, the results could be used to show the novelty of their data, instead of ours.

Actions:

1. We add more details to explain the biological relevance of the mapping rates at lines 185-189; now this part reads: “A high mapping rate indicates that majority of the microbial genomes in a sample could be represented by the target dataset, while a lower mapping rate indicates novel genomes in the sample that are not covered by the target dataset. Such a strategy has been extensively used in metagenomic analyses (PMID: 30661755, 31375809 and 32051016).”.
 2. As the reviewer suggested, we replaced the wrongly cited reference 37 with the correct one (PMID: 21508958).
-

Comment #4

Regarding the analysis shown in figure 2A, can the authors explain why the Firmicutes split into three clades? What's the source of the split? Could this split stem from the annotation tool (GTDB)?

Response:

Yes, GTDB splits Firmicutes into four clades, namely Firmicutes_A, Firmicutes_B, Firmicutes_C and Firmicutes. We checked manually and found that they all belong to Firmicutes according to NCBI taxonomy. We thus combined them into a single clade named ‘Firmicutes_all’.

Action:

We explained this at lines 305-306; now this part reads: “Firmicutes_all, including Firmicutes, Firmicutes_A, Firmicutes_B, and Firmicutes_C according to GTDB.”.

Comment #5

Figure 2A has no merit in the field of rumen and gut microbiome. I understand the importance of introducing your data, but this should not be displayed as the main figure and better suited for the supplementary data.

Response:

We thank the reviewer for his/her suggestion. In this study, Figure 2A is a key figure that summarizes the following information: 1) the taxonomic annotation our MAGs and their relative distributions in different phyla, for example, from this figure it is evident the Firmicutes_A represents the largest phyla, followed by Bacgeroidota, and then by others. 2) the annotation rates at different levels, and more importantly their uneven distributions on the phylogenetic tree. For example, we used a ‘star symbol’ to represent taxa not identified by GTDB (likely novel taxa). We can see from the tree that most of the species were not identified by GTDB; however, majority of the species could be assigned to known genera. Interestingly, the distributions of the unidentified species were not even to the genera, that is, some genera contained species that were all identified by GTDB (those without the stars; for example, the first Bacteroidota genus next to Firmicutes_A), while others contained mostly unidentified species. These results suggest that not only that GTDB did not cover most of the buffalo DT genomes at the species level, the ones that did include were strongly biased towards certain genera.

We thus believe Figure 2A is informative and indispensable.

Comment #6

The term "stomach", which is widely used by the authors, is not accurate. The reticulorumen is not a stomach. The actual "stomach" is only the abomasum. "stomach" should be replaced by the "upper digestive" tract or "foregut".

Response:

We thank the reviewer for the reminder. We agree that the term “stomach” is indeed inaccurate. However, the terms “upper digestive tract” and “foregut” are not suitable, too. Because they represent only the sites from the esophagus to the duodenum, not all the four stomach chambers. After consulted with the experts and surveyed the literature, including Chen *et al* (2019) and Stewart *et al* (2019) (PMIDs: 31221828, 31375809), we finally adopted the term “four-chambered stomach” and used it throughout the manuscript.

Action:

We replaced the term “stomach” with “four-chambered stomach” (or FC stomach for short) throughout our manuscript, including the main texts, materials and methods, figures and tables, and all supplementary materials.

Comment #7

Figure 3 begins with an excellent representation of MAGs distribution across the GIT. The alpha diversity measurement analysis displayed afterward is not innovative enough to be in the central figure. I understand the necessity to show the intra-sample diversity in each compartment and sub-compartment, but this analysis has no innovation. The authors did not try to "develop" this analysis into any downstream biological questions that may arise from their findings. Later on, the authors performed the requested beta diversity analysis. Despite its importance, this analysis is not innovative and has been widely presented in many GIT studies in multiple different hosts. Furthermore, can the authors relate to the horseshoe effect here? It is important to look at geographical locations that might explain the sparse clustering seen here, age as well.

Response:

We thank the reviewer for his/her comment. We agree with the reviewer that although alpha- and beta- diversity analyses were often shown in many literatures, the differences among sections and sites might not be unexpected. We thus moved them (i.e., the panels b, c and d) to supplementary.

As the reviewer suggested, we investigated the effects of host and environmental factors on the microbiota compositions. We used R package 'vegan' for PERMANOVA (Permutational multivariate analysis of variance) for single- and multiple-factor

analysis. We found that all factors exerted significant effects (Figure 3b), with DT sections being the strongest, followed by geography, breed, growth phase, and sex. We summarized these results into a new panel of Figure 3 (the new Fig.3b).

Actions:

1. As the reviewer suggested, we moved panels of b, c and d of Figure 3 to Supplementary Fig.5 and Supplementary Fig.6;
2. We performed PERMANOVA analysis and revealed significant effects of host and environmental factors on the microbiota compositions. We added a new Figure 3b to summarize these results. We also added the corresponding methods to lines 634-641; these new lines read: “PERMANOVA (Permutational multivariate analysis of variance) implemented in R package 'vegan' was used to investigate the effects of host and environmental factors on the microbiota compositions such as geography, DT sections, developmental stages, breed and sex. PERMANOVA is a permutation-based extension of multivariate analysis of variance to a matrix of pairwise distances, partitions within-group and between-group distances to permit assessment of the effect of an exposure or intervention (grouping factor) upon the sampled microbiome¹¹³. We performed both single- (i.e., to consider one factor at a time) and multi-factor (i.e., to consider all factors at the same time) analyses.”;
3. We also described and discussed the results in the main text at lines 292-301; now this part reads: “In order to maximize the discovery of DT associated microbial genomes, we collected samples from animals that varied in geographical locations, breeds, sexes and developmental stages, which were known to also affect microbial compositions. We thus also investigated the effects of these host and environmental factors on the microbiota using PERMANOVA (Permutational multivariate analysis of variance) implemented in R package 'vegan'. As we expected, in both single- and multi-factor analyses, DT sections exerted the strongest effects, followed by geography, breed, developmental stage, and sex (Fig.3b). Since the sampling strategy of this study was optimized to comparing microbial compositions across DT sections, we focused on the comparative analysis between DT sections, and left the comparisons of other factors for future studies.”.

Comment #8

The beautifully displayed per taxa sample analysis performed in figure 3E shows evident variation within a single body part. Why does it exist? Could other elements like geography, age, and diet have an effect here? As previously mentioned, the authors did not perform any downstream analysis and did not develop the biological question.

Response:

We thank the reviewer for this excellent suggestion. We agree with the reviewer that the variations could indeed be affected by host and environmental factors. We therefore first studied the effect of multiple factors on the microbial compositions using PERMANOVA analysis (Permutational multivariate analysis of variance). We found that the DT sections exerted the strongest effects, consistent with our expectations. See our response to Comment #7 for details.

All other factors also exerted significant effects, including geography, breed, age and sex (see the newly added Figure 3b). Since the sampling strategy of this study was optimized to compare microbial compositions across DT sections, we focused on the comparative analysis between DT sections, and left the comparisons of other factors for future studies.

Comment #9

The analysis presented in figure 4 shows no innovation and is widely performed. The authors did not elaborate on how they decided to focus on the four prokaryotes (Fig.4 e-h). Why did the authors determine that these are important to look at? Was it based on previous knowledge? I see no statistical test showing if the selected species have a higher correlation with archaea than other randomly chosen species. Other species also have the functions related upstream functions to methane production. I am missing the biological explanation for the choice. Furthermore, a quick web search will reveal that such analysis has been performed before.

Response:

We would like to apologize for not being able to explain our logic clearly. We first identified differentially abundant taxa in an unbiased manner using LEfSe (Supplementary Table 4), in other words, those showed significant enrichment in at least one DT sections as compared with others were chosen. We then focused on selected taxa in the main text based on the following logic: first, we selected taxa that were known to play important roles related to the physiological functions of the DT sections. These included *Fibrobacter*, *Ruminococcus*, and *Butyrivibrio* that contained the main microbes to degrade cellulose, and *Prevotella* that contained the main microbes to degrade hemicellulose. They happened to be enriched in the four stomach chambers. We thus also included *Butyrivibrio*, which showed similar patterns. Although similar analysis has been performed in previous studies, they focused only in rumen but not the whole DT. The innovation of our study is to include other DT sites, and revealed their distinct distributions along the DT; some of the distributions were actually unexpected, for example, contrary to previous beliefs, we *Fibrobacter* and *Archaea* were most abundant in omasum rather than in rumen.

We do agree with the reviewer that taxa enriched in other sections should be mentioned. We thus added a supplementary Figure 12 to include all marker taxa, included a brief introduction to them in the main text and discussed their putative functions in relations to their enriched sites.

Regarding the correlation between archaea and *Fibrobacter_all*, we do agree with the reviewer that an unbiased analysis should be performed. We thus calculated the correlations between archaea and all other major taxa at phylum and genus levels. We found that 164 genera are significantly related to Archaea, of which 88 genera were significantly positively related to Archaea, only 13 genera have functional information, and *Fibrobacter_all* was the most significant genus among them. We thus included these new results at lines 344-346.

Actions:

1. We added a new Supplementary Fig.9 to include all other marker taxa.
2. We revised the manuscript to better explain the whole process to identify marker taxa, and our logic for the selected ones in main text. These new contents could be found at lines 643-655; this part now reads “To identify microbes that show significantly differential abundances between sample groups of interests, we first calculated the relative redundances of the

MAGs in all samples. We mapped the clean reads of a sample to all MAGs and calculated the relative abundance of a MAG as the percentage of reads mapped to the MAG out of the total reads mapped to all MAGs in the sample; thus, the total relative abundances of all MAGs in sample are always 100%. Relative abundances for higher taxonomic levels such as genus, family and order were also determined by summing up the abundances of their daughter clades, according the phylogenetic tree provided by GTDB-TK. Linear Discriminant Analysis (LDA) implemented in the LEfSe tool was used to identify differential taxa between groups of samples. We chose those with LDA scores > 2 as the differential taxa. Wilcoxon test was used to show the statistical significance in the relative abundances of the differential taxa between groups. Then we selected taxa that are known to play an important role in physiological functions related to the grouping context.”.

3. We added a brief introduction to marker taxa that were enriched in other two sections and discussed their putative functions. These new contents are at lines 353-358. This part reads “In total, we identified 359 differentially abundant taxa at the genus level using LEfSe (Linear Discriminant Analysis (LDA) scores > 2). Among which, 172, 42 and 145 showed significant higher abundances in FC-stomach, intestine and rectum, respectively (Supplementary Table 4; Supplementary Fig.9). In addition to those mentioned above, the other differentially abundant taxa may also contribute to the physiological functions of the corresponding DT sections and should be investigated further.”.

4. As the reviewer suggested, we performed an unbiased analysis to identify taxa that correlated significantly with archaea. We added a brief description of the methods to lines 657-662; this part now reads “In order to identify the genus positively related to archaea, we calculated the correlations between archaea and all other major taxa at phylum and genus levels. All the genera of the Archaea phylum were combined with relative abundance, and the genus positively related to the Archaea phylum was selected based on the correlation significance in all the relevant results. Then according to the information search of the genus, manually remove the genus whose

function information cannot be retrieved”. We then added the results to lines 344-346; now this part reads “In addition, we performed correlation analysis between all bacterial genera and Archaea (Supplementary Table 3;), and found that Fibrobacter_all showed the highest correlation efficiency with the methanogens (Supplementary Fig.8).”.

Comment #10

Functional annotation analysis: Please elaborate more on your ORF\protein prediction process. I did not understand whether the COG was done in a supervised manner. Furthermore, was your process blind to sequences not framed by an ORF? I want to mention again that the analysis itself has been performed on buffalos in a previous study cited by the authors (Zhang et al. 2017). The authors did not seem to extend their analysis beyond the scope of the introduction.

Response:

We apologize for the misunderstanding. If we understand correctly, the reviewer thought we constructed orthologous groups using the predicted proteins from buffalo MAGs. We did not. In this study, we used an eggNOG-mapper tool to annotate the protein coding genes from the buffalo MAGs; eggNOG-mapper searched the protein sequences against a pre-compiled protein database and assigned our proteins to COG categories. It also assigned our proteins to KEGG pathways, among other things. More details can be seen in the “Methods” of our manuscript. See also <http://eggnog-mapper.embl.de/> for details about eggNOG-mapper.

Regarding the study by Zhang *et al* (2017), the authors sequenced ten buffalo fecal samples and obtained in total 3,172,144 non-redundant genes. In our study, we sequenced 695 samples from all DT sections; we obtained 4960 MAGs and 5,789,172 non-redundant genes. Thus, our dataset is far more comprehensive than that of Zhang *et al*.

Comment #11

Comparing their dataset to cattle data is problematic as cattle themselves are very varied in dietary composition, purpose (dairy versus meat cattle), and breed.

Did your analysis correct for these variates? Please be more specific regarding the nature of this comparison.

Response:

We agree with the reviewer and wish to thank him/her for this comment. Other factors like breed and diet indeed had a very important effect to the microbiome composition; however, since the metadata of cattle samples from Stewart *et al* were not available, we were not able to correct for these variates. We thus revised our interpretations of our findings; instead of attributing the differences between cattle and buffalo, we offered several alternative explanations including possible differences diets.

Action:

We revised the manuscript to offer alternative explanations to the different microbial compositions between cattle and buffalo rumens. The revised part can be found at lines 444-447, which reads “We would like to point out that the differences observed here were likely due to different diets between buffalo and cattle. However due to limited diet information from the public dataset of cattle, we were unable to dissect the effects of diets from other factors on the different microbial compositions. Future studies are thus needed for further illustration.”.

Comment #12

Last analysis- The authors should notice that this comparison is rather generalistic and that the abundance of these bacteria could vary in different cattle breeds. I am not sure that the prevotella utilize lactate. Please check and correct accordingly if needed.

Response:

We agree with the reviewer that we should not generalize our interpretations. As we mentioned in response to Comment #11, we revised this part to discuss briefly the possible roles of diets.

As the reviewer requested, we checked the literatures and found some *Prevotella* strains including *P. albensis* and *P. bryantii*) could utilize lactate.

Actions:

1. We revised our interpretations regarding the differences between cattle and buffalo rumens. The revised part can be found at lines 430-431, which reads “Bacteroidota species, especially those in its dominant genus *Prevotella* could utilize lactate and capable of degrading non-cellulose plant fibers.”.
 2. We cited the publication that showed *Prevotella* strains could utilize lactate (PMIDs: 33339094).
-
-

Response to referee 2

NOTE: for convenience purposes, we broke the reviewers’ comments into relevant sections, numbered them and provided point-to-point responses. We did not truncate nor rephrase the comments by any means.

General remark

The authors present an interesting and comprehensive analysis of the buffalo digestive tract from a metagenomics perspective.

Response:

We thank the reviewer for his/her valuable suggestion that have helped to improve this manuscript, and his/her positive opinions on our research.

Comments for the Author

Comment #1

line 89 misses some references:

Cow/cattle:

<https://www.nature.com/articles/s41587-019-0202-3>

<https://www.nature.com/articles/s41598-021-81668-9>

<https://genomebiology.biomedcentral.com/articles/10.1186/s13059-020-02144-7>

<https://www.biorxiv.org/content/10.1101/2021.04.02.438222v1>

Reference 36 says sheep but the paper is from goat

Chickens:

<https://genomebiology.biomedcentral.com/articles/10.1186/s13059-020-1947-1>

<https://peerj.com/articles/10941/>

Response:

We thank the reviewer for reminding us. We have added them to appropriated sections of our manuscript.

Actions:

1. We added these references to appropriated sections of our manuscript.
 2. We corrected reference 36 from “sheep” to “goat” at line 85.
-

Comment #2

line 116 - were the reads paired-end?

Response:

Yes, the sequencing reads were paired-end. The details can be found in section “Quality control and removal host- and food-associated genomes” of the Methods lines 561-562 of the revised manuscript; now this part reads “Paired-end raw sequencing reads were first trimmed by Trimmomatic (v.0.35) to remove vectors and low-quality bases”.

Comment #3

Lines 157 and 159 have different reference formats?

Line 159 references cattle MAGs and reference 37, but 37 is not a valid reference here

Response:

We apologize for the mistake.

Actions:

1. We revised our references into a unified format.
2. We replaced the wrong citation here (ref 37) with the correct one (PMID: 21508958).

Comment #4

Line 161 and 162, high quality MAGS as defined by Bowers et al include full 16S gene and most tRNA genes - are these cut-offs included in the analysis here?

Response:

No. We doubled checked the literature, Bowers *et al* defined the “high quality MAGs” as “High-quality draft will indicate that a SAG or MAG is >90% complete with less than 5% contamination.” (PMID: 28787424). Full 16S gene and most tRNA genes are valid indicators for high quality MAGs, but they are not part of the definition. We used Bowers *et al*'s definition in our analysis.

Comment #5

Line 420 and elsewhere - "better" is not the correct word here! I suspect highly adapted western breeds of cattle are highly efficient at carbohydrate degradation of certain carbohydrates; the differences with buffalo will almost certainly be due to differences in diet

Response:

We do agree. As the reviewer suggested, we revised this part and attributed the microbial differences between cattle and buffalo rumens to diet instead of the animals.

Actions:

1. We revised this part and attributed the microbial differences between cattle and buffalo rumens to diet instead of the animals. This part can be found at lines 519-526, which reads “Microbial compositions were known to be affected by both intrinsic factors such as DT section, sex, breeds and developmental stages, as well as environmental factors such as geographical locations. In this study, we found all factors exerted significant effects on DT microbiota in buffalo in both single- and multi-factor analyses (see Materials and Methods); notably, DT sections exerted the strongest effects, followed by geography, breed, developmental stage, and sex. Since the sampling strategy of this study was optimized to comparing microbial compositions across DT sections, we focused on the comparative analysis between DT sections, and left the comparisons of other factors for future studies.”.

2. We change the word “better” to “stronger” at line 440.

Comment #6

*505 - sample collection - how were they sampled? Stomach tube? After slaughter?
Some other way?*

Response:

We apology for not showing the statistics in the manuscript. Out of the total 695 samples, 497 were taken from live buffalo, including 98 rumen samples collected by stomach tube and 399 fecal samples collected by disposable soft long arm gloves from rectum. The remaining 198 samples were all from slaughtered buffalo, including 31 rumen samples, 28 reticulum samples, 32 omasum samples, 22 abomasum samples, 24 jejunum samples, 31 cecum samples and 30 colon samples.

Action:

We added the above summary to the “sample collection” section at liens 538-543, now this part reads: “Out of the total 695 samples, 497 were taken from live buffalo, including 98 rumen samples collected by stomach tube and 399 fecal samples collected by disposable soft long arm gloves from rectum. The remaining 198 samples were all from slaughtered buffalo, including 31 rumen samples, 28 reticulum samples, 32 omasum samples, 22 abomasum samples, 24 jejunum samples, 31 cecum samples and 30 colon samples (Supplementary Table 1, 8).”.

Comment #7

Also how long were they stored in the freezer?

Response:

No more than a week. All samples were immediately frozen after collection in liquid nitrogen and stored at -80°C until DNA extraction. These samples were often processed within one or two days.

Action:

We added the following sentence to the manuscript at lines 547-548: “Most samples were submitted to DNA extraction within a week after collection.”.

Comment #8

lines 540 - 543 - this is an unusual way of using SPAdes which usually combines data from multiple kmers

Response:

According to the online manual, (meta)SPAdes will automatically select the k-mer sizes for graph construction (metaSPAdes user manual). Thus, by default each sample should get its optimized k-mer. In this study, we did the k-mer selection manually instead: we tested various k-mer and chose the one produced best N50.

Comment #9

549 - combining data from two different assemblers needs to be described more - how was this done?

Response:

We apology for the confusion. For each sample, we did the assembly using both metaSPAdes and MEGAHIT, and chose the one with better N50 value as the final assembly of this sample.

Action:

We added the above descriptions to the manuscript at lines 574-577, which reads “We compare the total length, N50 and the contigs numbers of the assembly using metaSPAdes and MEGAHIT for each sample, and select the assembly with a relatively long total length and N50 as the final assembly of this sample.”.

Comment #10

Line 559 when using dRep, the authors should state and justify which overlap threshold they used (-nc)

Response:

We thank the reviewer for the reminders. In our study, the default (-nc 0.1) parameter for overlap threshold was used.

Action:

As the reviewer suggested, we explained this and added the default (-nc 0.1) parameter of overlap threshold in the “Methods” at lines 119-120.

Comment #11

Data availability - I cannot assess the public data as the authors have not made it available.

Response:

We apology for the inconvenience. The datasets were scheduled to be public after the manuscript was accepted. As the reviewer requested, we have now made public of the raw sequencing data and the sequences of the 4,960 MAGs. They are available at the NCBI SRA database under the accession ID [PRJNA656389](https://www.ncbi.nlm.nih.gov/bioproject/PRJNA656389) (<https://www.ncbi.nlm.nih.gov/bioproject/PRJNA656389>) and the ENA under the accession ID [PRJEB43196](https://www.ebi.ac.uk/ena/browser/view/PRJEB43196) (<https://www.ebi.ac.uk/ena/browser/view/PRJEB43196>), respectively.

REVIEWER COMMENTS

Reviewer #1 (Remarks to the Author):

I am afraid I have to firmly disagree with the authors in this case. As comprehensive their profiling of the buffalo may be and despite its large potential, a mere data description is not enough these days. The authors compare their study to the draft human genome project, yet they neglect the fact that this project originated well in the 20th century (launched in 1990 according to Wikipedia), over 30 years ago. These days it is expected that authors will go beyond their ability to simply describe the data. The authors took great pride in performing this study on samples from buffalos that varied in geography, breed, developmental stage, and sex. The study of the differences within and between these variables would be of great interest. For example, the authors could have explored the effect of geography on function distribution. Despite the fact that previously published buffalo studies are significantly smaller than this study, the overall notation presented in both this study and the previous one remains largely the same. Lastly, I would like to thank the authors for thoroughly addressing each comment, I recommend that the authors should address their work to a different journal with more relevance to microbial ecology.

Reviewer #2 (Remarks to the Author):

Many thanks to the authors who have answered some of my comments satisfactorily, but not all. Specifically:

Comment #4

The authors are wrong

Bowers et al is unequivocal in Table 1

<https://www.nature.com/articles/nbt.3893/tables/1>

High-quality draft (SAG/MAG)

Multiple fragments where gaps span repetitive regions. Presence of the 23S, 16S, and 5S rRNA genes and at least 18 tRNAs.

Comment #8

N50 is not a good quality metric for metagenomics assemblies, it is only a good metric if we know the final genome size, and in metagenomics we don't know that.

One could, for instance, optimise the total amount of sequence assembled, or the total amount assembled in contigs > 1kb etc

An assembly with two contigs of 100kb would have a high N50, but only 200kb of sequence

An assembly with 1,000,000 contigs and an average size of 5kb would have a lower N50, but would arguably be a better assembly because it contains more data

I think this is an unusual use of metaSPAdes but I accept you cannot go back and change this. I think more description and justification of the choice needs to be given

Comment #9

See above why N50 is a poor metric to optimise in metagenomics

Comment #10

Using the default overlap threshold is a little odd as it requires the ANI threshold to only apply over 10% of the genome

If you look at other papers in the field, they alter the `-nc` option to fit their data, depending on level of completeness:

<https://www.nature.com/articles/s41587-020-0603-3> use `-nc 0.3`

<https://www.nature.com/articles/s41586-019-0965-1> use `-nc 0.6`

I think the dRep analysis needs to be re-run with this parameter adjusted

Comment #11

I thank the authors for sharing their data :)

I think "gut metagenome[Taxonomy ID: 749906]" is not the best taxonomic category to use here, I would prefer something more specific such as "bovine gut metagenome"

The ENA data does not appear to be present

The NCBI SRA data has raw data but not MAGs

Response to referee 1

NOTE: for convenience purposes, we broke the reviewers' comments into relevant sections, numbered them, and provided point-to-point responses. We did not truncate nor rephrase the comments by any means.

General remark

I am afraid I have to firmly disagree with the authors in this case. As comprehensive their

profiling of the buffalo may be and despite its large potential, a mere data description is not enough these days. The authors compare their study to the draft human genome project, yet they neglect the fact that this project originated well in the 20th century (launched in 1990 according to Wikipedia), over 30 years ago. These days it is expected that authors will go beyond their ability to simply describe the data. The authors took great pride in performing this study on samples from buffaloes that varied in geography, breed, developmental stage, and sex. The study of the differences within and between these variables would be of great interest. For example, the authors could have explored the effect of geography on function distribution. Despite the fact that previously published buffalo studies are significantly smaller than this study, the overall notation presented in both this study and the previous one remains largely the same. Lastly, I would like to thank the authors for thoroughly addressing each comment, I recommend that the authors should address their work to a different journal with more relevance to microbial ecology.

Response:

We thank you for your evaluation of our study and apologize for not explaining the novelty of our study clearly. Please allow us to explain them one more time, as detailed below.

First, we provided the largest and most comprehensive catalog of microbial genomes associated with the digestive tract (DT) of buffalo. Our catalog consists of 4,960 high-quality metagenome-assembled genomes (MAGs), obtained using 695 samples from eight sites along the DT of buffalo. We confirmed with rarefaction analysis (see Figure 1B) that our MAG set could represent most of the DT-associated microbes in buffalo because the number of MAGs did not increase with more samples. We also annotated 5,862,748 non-redundant proteins, a significant proportion of which was not found in public databases.

In comparison, **our study has overcome the limitations presented in previous studies on buffalo metagenomics.** For example, Henderson and colleagues surveyed rumen microbiota of 32 animals including buffalo using 16S sequencing results [1]. Although they collected a large number of samples (e.g., 742) and studied the effects of geographical locations on rumen microbiota, their conclusions are limited by the low resolution of 16S sequencing for taxonomic classification, and no functional characterization. In addition, only rumen samples were surveyed in this study. In another study, Zhang and colleagues compared the gut microbiome between buffalo and cattle [2],

but they only sampled ten buffaloes. Thus, previous studies either used the amplicon sequencing method or had a very limited sample size; in addition, they focused on a particular section of the GIT (rumen or intestinal).

We wish to emphasize here that the sample size does matter in metagenomic studies, especially in identifying buffalo-associated microbes. As shown in Figure 1A, we show that the number of unique MAGs increases with the increasing number of samples; thus, studies with 100 samples or less may cover only a small proportion of microbes, and lead to biased results and conclusions.

Second, our study went beyond simple “data description”. In addition to genome assembly and annotation, we identify microbes that showed differentially abundant microbes between DT sections and/or sites in an unbiased manner, and the effects on functional differences. In addition, we also revealed that other factors, including geography, gender, and development stage contributed significantly to the variation of DT microbiota. As shown in Figure 3B, DT sections exerted the strongest effects, followed by geography, breed, developmental stage, and sex. Since the sampling strategy of this study was optimized to comparing microbial compositions across DT sections, **we focused on the comparative analysis between DT sections and left the comparisons of other factors (such as geography) for future studies.**

To summarize, our study sampled from eight sites of buffalo's digestive tract (DT), and provided the largest and most comprehensive catalogs of microbial genomes and genes associated with buffalo; our study overcame the limitations of previous studies in buffalo DT metagenomes and provided researchers with an unbiased view of the buffalo's MAGs. Our study also provided insights into taxonomical and functional associations between DT metagenomes and different sites, warrant further and more detailed investigation.

The “human genome” analogy was used to illustrate the importance of the first comprehensive profiling of a (meta)genome in promoting the development of the related field. We hope that with our resource and preliminary analysis results, we can facilitate future research on the functions of the microbes and their roles in the different DT sites, and intervention for better buffalo's quality through DT microbes.

Action:

- We cited the two references mentioned above as refs. 25 and 35 respectively, and briefly summarised their limitations (page 4, lines 82-83 and 89-90).

References

1. Henderson, G., et al., *Rumen microbial community composition varies with diet and host, but a core microbiome is found across a wide geographical range*. *Sci Rep*, 2015. **5**: p. 14567.
2. Zhang, J., et al., *Comparative study of the gut microbiome potentially related to milk protein in Murrah buffaloes (*Bubalus bubalis*) and Chinese Holstein cattle*. *Sci Rep*, 2017. **7**: p. 42189.

Response to referee 2

General remark

Many thanks to the authors who have answered some of my comments satisfactorily, but not all.

Comments for the Author

Comment #4

The authors are wrong

Bowers et al is unequivocal in Table 1 <https://www.nature.com/articles/nbt.3893/tables/1>

High-quality draft (SAG/MAG)

Multiple fragments where gaps span repetitive regions. Presence of the 23S, 16S, and 5S rRNA genes and at least 18 tRNAs.

Response:

We thank you for pointing this out and apologize for the omission.

Actions:

1. As you suggested, we re-analyzed our data using the criteria defined by Bowers *et al* in their NBT paper, and identified 1,575 (31.75%) MAGs as high-quality draft genomes, i.e., these MAGs were at least 90% complete, with less than 5% of contamination and had 23S, 16S and 5S rRNA genes and at least 18 tRNAs.
2. We revised the manuscript accordingly (page 9, lines 184-186);

3. We revised the “Methods” section of the manuscript on page 35, lines 890-892;
4. We also revised Figure 1c&d.

Comment #8

N50 is not a good quality metric for metagenomics assemblies, it is only a good metric if we know the final genome size, and in metagenomics we don't know that.

One could, for instance, optimise the total amount of sequence assembled, or the total amount assembled in contigs > 1kb etc.

An assembly with two contigs of 100kb would have a high N50, but only 200kb of sequence.

An assembly with 1,000,000 contigs and an average size of 5kb would have a lower N50, but would arguably be a better assembly because it contains more data.

I think this is an unusual use of metaSPAdes but I accept you cannot go back and change this.

I think more description and justification of the choice needs to be given.

Response:

We thank you for sharing your insights with us and your valuable suggestions.

As you suggested, we tested the effects of the quality metrics (N50 vs. mapping rate) on the number of reads assembled. Due to time constraints, we randomly selected 16 samples and re-assembled them using metaSPAdes with seven different k-mers (from 21 to 141 with a step size of 20). We then mapped the reads to the assembled scaffolds (>1kb) for each k-mer, and chose the k-mer with the highest mapping rate for subsequent analyses (referred as to "Kmer_MRate"). For comparison, we also mapped the reads to the scaffolds (>1kb) we obtained to maximize the N50 (referred as to "Kmer_N50"). In comparison, we also did binning using the generated scaffolds for each sample and calculated the mapped rate of the reads to the bins.

As shown in the Figure below, we found that at the level of the scaffold, the mapping rates of the Kmer_MRate group were significantly higher than the Kmer_N50 group; this is expected because the Kmer_Mrate group was optimized for higher mapping rates. **However, we did not find a significant difference at the bins level between the two groups.**

Because modern assemblers including metaSPAdes did not report the number of assembled reads, the reads mapping took us on average additional 60 CPU hours per sample. Therefore, we used the Kmer_N50 results instead of Kmer_MRate for subsequent analyses.

Effects of different assembly parameters on reads mapping rates to obtained scaffolds (left panel) and bins (right panel). Each dot represents a sample. **Left panel:** at the level of the scaffold, the mapping rates of the Kmer_MRate group were significantly higher than the Kmer_N50 group; **Right panel:** after binning, we found no significant difference at the bins level between the two groups. Kmer_N50: the Kmer used in our study to obtain the highest N50; Kmer_MRate: the Kmer with the highest mapping rate (range 55 ~ 115, step increase 10). T-test was used to compare between groups. Level of significance: *** $P < 0.001$; ** $P < 0.01$; * $P < 0.05$; NS. $P \geq 0.05$.

Actions:

1. We added a brief description of the above analysis and results to the “Methods” section of the manuscript on page 32-33, lines 667-818.
2. We added the above Figure as Supplementary Fig.12.
3. We also added the detailed mapping rates of the 16 selected samples as Supplementary Table 11 to the manuscript.

Comment #9

See above why N50 is a poor metric to optimize in metagenomics

Response:

Please see our responses and action points to Comment #8.

Comment #10

Using the default overlap threshold is a little odd as it requires the ANI threshold to only apply over 10% of the genome.

If you look at other papers in the field, they alter the -nc option to fit their data, depending on level of completeness:

<https://www.nature.com/articles/s41587-020-0603-3> use -nc 0.3

<https://www.nature.com/articles/s41586-019-0965-1> use -nc 0.6

I think the dRep analysis needs to be re-run with this parameter adjusted.

Response:

We agree with you and have tested four different options of the ‘-nc’ parameter (i.e., -nc = 0.1, 0.2, 0.3 or 0.6) in selected samples.

As shown in the table below, we selected four groups of samples to test the ‘-nc’ parameter. For each group, we obtained the same number of de-replicated bins regardless of the ‘-nc’ options. We also combined the bins from all samples and did the de-replication, and again obtained the same number of de-replicated bins.

De-replication (dRep) results using different ‘-nc’ options

Source (samples)	bins (input) *	dRep-ed bins							
		-nc 0.1**		-nc 0.2		-nc 0.3		-nc 0.6	
		bins (output)	bins (output; combined)	bins (output)	bins (output; combined)	bins (output)	bins (output; combined)	bins (output)	bins (output; combined)
Rumen set 1 (18)	1241	203	1022	203	1022	203	1022	203	1022

Abomasum Cecum Colon (83)	2994	457		457		457		457
Faeces (31)	1922	151		151		151		151
Rumen set 2 (62)	1786	236		236		236		236

* total number of bins obtained from MetaBAT2;

** the parameter used in this study.

We then confirmed that the scaffolds obtained with different ‘-nc’ options were the same using GTDB-TK, with only one exception. We manually checked this bin, and found it belonged to the same genus according to GTDB-TK, as shown below.

Together, these results suggest that different values of the ‘-nc’ do not have significant effects on our results. We thus used ‘-nc 0.1’ in this study.

Taxonomic annotation of the different bins.

GTDB-TK			
dRep_nc	bins	species	genus
0.1*	S-24_FDME192071042-1a_bin.32	N/A	d__Bacteria;p__Bacteroidota;c__Bacteroidia;o__Bacteroidales;f__P3;g__Phil12
0.2	S-516_FDME192071037-1a_bin.8	N/A	d__Bacteria;p__Bacteroidota;c__Bacteroidia;o__Bacteroidales;f__P3;g__Phil12
0.3	S-516_FDME192071037-1a_bin.8	N/A	d__Bacteria;p__Bacteroidota;c__Bacteroidia;o__Bacteroidales;f__P3;g__Phil12
0.6	S-516_FDME192071037-1a_bin.8	N/A	d__Bacteria;p__Bacteroidota;c__Bacteroidia;o__Bacteroidales;f__P3;g__Phil12

* the parameter used in this study.

Actions:

1. We summarized the above results in the “Methods” section on page 33-34, lines 837-848;
2. We added the above two tables as Supplementary Table 12 and Supplementary Table 13 to the manuscript.

Comment #11

I thank the authors for sharing their data :)

I think "gut metagenome[Taxonomy ID: 749906]" is not the best taxonomic category to use here,

I would prefer something more specific such as "bovine gut metagenome".

The ENA data does not appear to be present.

The NCBI SRA data has raw data but not MAGs.

Response:

We apologize for the omission. The ENA address we obtained did not seem to be able to directly access the MAGs data.

Actions:

- 1.** We resubmitted the assembled MAGs to ENA. The data is now accessible at https://www.ebi.ac.uk/ena/browser/view/GCA_905249985.1.
- 2.** We updated the manuscript with this new URL on page 37-38, lines 940-946.

REVIEWER COMMENTS

Reviewer #2 (Remarks to the Author):

The authors compared a normal use of metaSPAdes vs their particular use of metaSPAdes in terms of mapping rate

Mapping rate is shown to be better to the normal assembly than the author's chosen assembly, but no difference to the bins

However, would the bins not be better from the normal use of metaSPAdes? This should be tested. Do we get more and better bins from the normal use of metaSPAdes?

In terms of the dRep analysis, I repeated the author's analysis and found approximately the same results.

However, I noticed that the MAGs had only been de-replicated at strain level (-sa 0.99) and not species level (-sa 0.95). The authors should also de-replicate at species level and present the results.

The data submission is also inadequate. The authors present all 4000+ MAGs as a single FASTA file and one must download that FASTA file and parse the FASTA headers to recreate the bins. Even worse, this single FASTA is annotated as coming from *Bubalus bubalis* (water buffalo) which makes it impossible to use for others. ENA has a process for submitting MAGs and this should be followed: <https://ena-docs.readthedocs.io/en/latest/submit/assembly/metagenome/mag.html>. Each MAG will need to be assigned the correct taxonomy

Response to referee 2

NOTE: for convenience purposes, we broke the referee's comments into relevant sections, numbered them, and provided point-to-point responses. We did not truncate nor rephrase the comments by any means.

Comment #1

The authors compared a normal use of metaSPAdes vs their particular use of metaSPAdes in terms of mapping rate.

Mapping rate is shown to be better to the normal assembly than the author's chosen assembly, but no difference to the bins.

However, would the bins not be better from the normal use of metaSPAdes? This should be tested. Do we get more and better bins from the normal use of metaSPAdes?

Response:

We thank you very much for this relevant and valuable suggestion.

As you suggested, we tested the effects of the quality metrics (Kmer_MRate vs. Kmer_N50) on the assembled bins at strain and species levels respectively (drep -sa = 0.99 and 0.95). Using the same test dataset as our last revision, at the strain level, we did obtain more MAGs using the Kmer_MRate method ("the normal use"); however, the Kmer_MRate method generated significantly fewer high-quality MAGs (69 vs. 85 of the Kmer_N50 method; see Fig 1a below). Additionally, the MAGs generated by Kmer_MRate have significantly shorter N50s (Fig 1b) and genome lengths (Fig 1c), and are significantly fragmented (Fig 1d). We found similar results at the species level (i.e., MAGs are dereplicated using -sa=0.95; Fig. 1e-h).

Together, the Kmer_MRate method generated more MAGs at the cost of genome size and quality. We thus used the Kmer_N50 method in our manuscript.

Fig 1. Quality assessment (i.e., completeness and contamination statistics) of the non-redundant MAGs generated using Kmer_MRate and Kmer_N50 methods. a) Pie chart shows the numbers and relative proportions of the high-, medium- and low-quality MAGs generated using the Kmer_MRate and Kmer_N50 groups at the strain level. Here the “high-quality” MAGs are defined as those with $\geq 90\%$ completeness and $\leq 5\%$ contamination, and presence of the 23S, 16S, and 5S rRNA genes and at least 18 tRNAs. All other MAGs are $>80\%$ complete and $\leq 10\%$ contaminated. Those in blue have a quality score ≥ 50 as defined by Parks *et al*¹, whereas those in grey have a quality score <50 . b), c) and d) show the comparisons of N50, genome size and the number of contig per genome respectively for the MAGs assembled at strain level between the Kmer_MRate and Kmer_N50 methods. Wilcoxon rank sum test was used to compare between groups. Level of significance: *** $P < 0.001$; ** $P < 0.01$; * $P < 0.05$; NS. $P \geq 0.05$. e) Pie chart shows the numbers and relative proportions of the high-, medium- and low-quality MAGs generated by the two at the species level. f), g) and h) show the comparisons of N50, genome size and the number of contig per genome respectively for the MAGs assembled at species level between the two methods.

Actions:

1. We added a brief description of the above analysis and results to the “Methods” section of the manuscript on page 34-35, lines 614-625.
2. We added the above Figure as Supplementary Fig.14.

Comment #2

In terms of the dRep analysis, I repeated the author's analysis and found approximately the same results.

However, I noticed that the MAGs had only been de-replicated at strain level (-sa 0.99) and not species level (-sa 0.95). The authors should also de-replicate at species level and present the results.

Response:

Thank you for your suggestion.

As you suggested, we de-replicated all bins at the species level and obtained a non-

redundant set of 3,255 MAGs with completeness $\geq 80\%$ and contamination $\leq 10\%$. We further predicted the presence of the 23S, 16S, and 5S rRNA genes and tRNAs, and assigned taxonomic classifications to the MAGs using GTDB-TK (Fig. 2).

We emphasized in the manuscript that most results were based on strain-level analysis. In the future, we will analyze the species level data and put all results into an independent database.

Fig.2 Quality assessment and taxonomic classification of 3,255 species-level MAGs. a)

Completeness and contamination measurements of the 3,255 MAGs. Each point represents a MAG. Red points indicate the high-quality genomes with $\geq 90\%$ completeness, $\leq 5\%$ contamination, and presence of the 23S, 16S, and 5S rRNA genes and at least 18 tRNAs. All other MAGs are $>80\%$ complete and $\leq 10\%$ contaminated. Those in blue have a quality score ≥ 50 , whereas those in grey have a quality score <50 . b) Pie chart shows the numbers and relative proportions of the red, blue, and grey MAGs in c). Histograms in c), d) and e) show the distributions of N50, genome size, and the number of contig per genome respectively for the 3,255 MAGs. f) The classification rates of 3,255 MAGs at different taxonomic levels. The numbers above the pie charts indicate the percentages of MAGs (out of 3,255) that could be annotated at the respective levels; the numbers inside the pie charts indicate the percentages of archaea (orange) and bacteria (green) of each pie. g) The classification rates of archaea (left) and bacteria (right) at different taxonomic levels. The numbers indicate the amounts of MAGs classified.

Actions:

1. We summarized the above results in the “Results” section on page 10, lines 171-174; page 11, lines 193-196 and page 13, lines 233-238;
2. We added the above Figure as Supplementary Fig.2.

Comment #3

*The data submission is also inadequate. The authors present all 4000+ MAGs as a single FASTA file and one must download that FASTA file and parse the FASTA headers to recreate the bins. Even worse, this single FASTA is annotated as coming from *Bubalus bubalis* (water buffalo) which makes it impossible to use for others. ENA has a process for submitting MAGs and this should be followed: <https://ena-docs.readthedocs.io/en/latest/submit/assembly/metagenome/mag.html>. Each MAG will need to be assigned the correct taxonomy*

Response:

We apology for the inconvenience. We have resolved all issues related to the Figshare,

see our action points below. However, we still have some trouble with the ENA submission.

We have sent emails to the ENA helpdesk for help, but yet to hear from them; we will update the ENA submission as soon as we can fix the issues.

Actions:

1. As you requested, we resubmitted the data to Figshare. The sequences and annotations of the 4,960 strain-level MAGs are available at Figshare (https://figshare.com/articles/dataset/Buffalo_digestive_tract_metagenome_MAGs/17000302). Two files are available for download: 1) a 'Buffalo_DT_4960_MAGs.tar.gz' file that contains the sequences of the MAGs in fasta format, one MAG a file; 2) a text file 'MAGs_information.txt' contains detailed information of the MAGs including size, completeness, contamination and taxonomic assignments by GTDB. Please note that due to the large file size, the "Download all" link does not work; however, users can download the two individual files.
2. We also uploaded the annotations of the 3,255 species-level MAGs to Figshare (https://figshare.com/articles/dataset/Species-level_3255_Buffalo_digestive_tract_metagenome_MAGs/17081366). It also provides MAGs split by bins and the corresponding information files can be downloaded. Please note that due to the large file size, the "Download all" link does not work; however, users can download the two individual files, namely 'Buffalo_species_level_3255_MAGs.tar.gz' and 'Buffalo_species_level_3255_MAGs_information.tsv'.
3. We included the URLs to page 40, lines 734-739 in the revised manuscript.

Reference:

1. Parks DH, *et al.* Recovery of nearly 8,000 metagenome-assembled genomes substantially expands the tree of life (vol 2, pg 1533, 2017). *Nature Microbiology* **3**, 253-253 (2018).

REVIEWERS' COMMENTS

Reviewer #2 (Remarks to the Author):

I am happy that the authors have addressed all of my comments